# TANGOS: Regularizing Tabular Neural Networks through Gradient Orthogonalization and Specialization

**Alan Jeffares**[*]
University of Cambridge
aj659@cam.ac.uk

**Tennison Liu**[*]
University of Cambridge
tl522@cam.ac.uk

**Jonathan Crabbé**
University of Cambridge
jc2133@cam.ac.uk

**Fergus Imrie**
University of California, Los Angeles
imrie@ucla.edu

**Mihaela van der Schaar**
University of Cambridge
Alan Turing Institute
mv472@cam.ac.uk

## Abstract

Despite their success with unstructured data, deep neural networks are not yet a panacea for structured tabular data. In the tabular domain, their efficiency crucially relies on various forms of regularization to prevent overfitting and provide strong generalization performance. Existing regularization techniques include broad modelling decisions such as choice of architecture, loss functions, and optimization methods. In this work, we introduce Tabular Neural Gradient Orthogonalization and Specialization (TANGOS), a novel framework for regularization in the tabular setting built on latent unit attributions. The gradient attribution of an activation with respect to a given input feature suggests how the neuron *attends* to that feature, and is often employed to interpret the predictions of deep networks. In TANGOS, we take a different approach and incorporate neuron attributions directly into training to encourage orthogonalization and specialization of *latent attributions* in a fully-connected network. Our regularizer encourages neurons to focus on sparse, non-overlapping input features and results in a set of diverse and specialized latent units. In the tabular domain, we demonstrate that our approach can lead to improved out-of-sample generalization performance, outperforming other popular regularization methods. We provide insight into *why* our regularizer is effective and demonstrate that TANGOS can be applied jointly with existing methods to achieve even greater generalization performance.

## 1 Introduction

Despite its relative under-representation in deep learning research, tabular data is ubiquitous in many salient application areas including medicine, finance, climate science, and economics. Beyond raw performance gains, deep learning provides a number of promising advantages over non-neural methods including multi-modal learning, meta-learning, and certain interpretability methods, which we expand upon in depth in Appendix C. Additionally, it is a domain in which general-purpose regularizers are of particular importance. Unlike areas such as computer vision or natural language processing, architectures for tabular data generally do not exploit the inherent structure in the input features (i.e. locality in images and sequential text, respectively) and lack the resulting inductive biases in their design. Consequentially, improvement over non-neural ensemble methods has been less pervasive. Regularization methods that implicitly or explicitly encode inductive biases thus play a more significant role. Furthermore, adapting successful strategies from the ensemble literature to neural networks may provide a path to success in the tabular domain (e.g. Wen et al., 2020). Recent work in Kadra et al. (2021) has demonstrated that suitable regularization is essential to

---

[*]Equal contribution

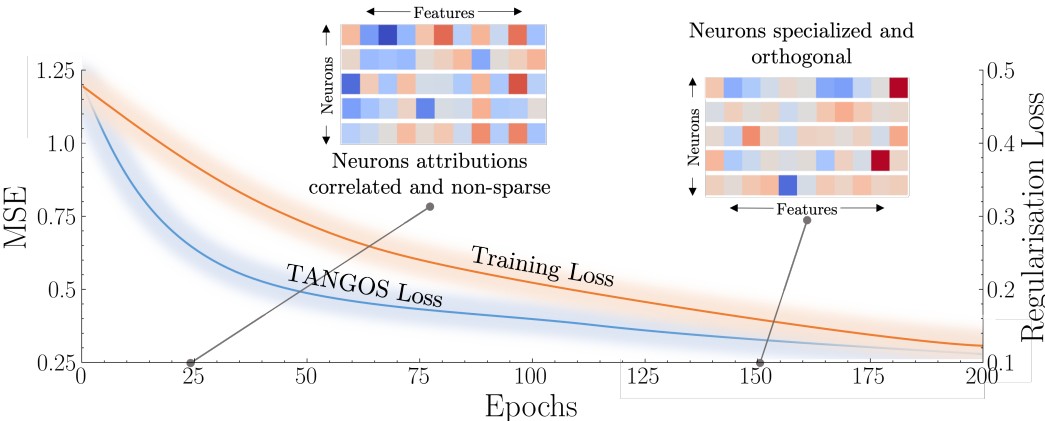

Figure 1: **TANGOS encourages specialization and orthogonalization.** `TANGOS` penalizes neuron attributions during training. Here, ■ indicates strong positive attribution and ■ indicates strong negative attribution, while interpolating colors reflect weaker attributions. Neurons are regularized to be *specialized* (attend to sparser features) and *orthogonal* (attend to non-overlapping features).

outperforming such methods and, furthermore, a balanced *cocktail* of regularizers results in neural network superiority.

Regularization methods employed in practice can be categorized into those that prevent overfitting through data augmentation (Krizhevsky et al., 2012; Zhang et al., 2018), network architecture choices (Hinton et al., 2012; Ioffe & Szegedy, 2015), and penalty terms that explicitly influence parameter learning (Hoerl & Kennard, 1970; Tibshirani, 1996; Jin et al., 2020), to name just a few. While all such methods are unified in attempting to improve out-of-sample generalization, this is often achieved in vastly different ways. For example, $L1$ and $L2$ penalties favor sparsity and shrinkage, respectively, on model weights, thus choosing more parsimonious solutions. Data perturbation techniques, on the other hand, encourage smoothness in the system assuming that small perturbations in the input should not result in large changes in the output. Which method works best for a given task is generally not known *a priori* and considering different classes of regularizer is recommended in practice. Furthermore, combining multiple forms of regularization simultaneously is often effective, especially in lower data regimes (see e.g. Brigato & Iocchi, 2021 and Hu et al., 2017).

Neuroscience research has suggested that neurons are both *selective* (Johnston & Dark, 1986) and have *limited capacity* (Cowan et al., 2005) in reacting to specific physiological stimuli. Specifically, neurons selectively choose to focus on a few chunks of information in the input stimulus. In deep learning, a similar concept, commonly described as a *receptive field*, is employed in convolutional layers (Luo et al., 2016). Here, each convolutional unit has multiple filters, and each filter is only sensitive to specialized features in a local region. The output of the filter will activate more strongly if the feature is present. This stands in contrast to fully-connected networks, where the all-to-all relationships between neurons mean each unit depends on the entire input to the network. We leverage this insight to propose a regularization method that can encourage artificial neurons to be more specialized and orthogonal to each other.

**Contributions. (1) Novel regularization method for deep tabular models.** In this work, we propose `TANGOS`, a novel method based on regularizing neuron attributions. A visual depiction is given in Figure 1. Specifically, each neuron is more *specialized*, attending to sparse input features while its attributions are more *orthogonal* to those of other neurons. In effect, different neurons pay attention to non-overlapping subsets of input features resulting in better generalization performance. We demonstrate that this novel regularization method results in excellent generalization performance on tabular data when compared to other popular regularizers. **(2) Distinct regularization objective.** We explore how `TANGOS` results in distinct emergent characteristics in the model weights. We further show that its improved performance is linked to increased diversity among weak learners in an ensemble of latent units, which is generally in contrast to existing regularizers. **(3) Combination with other regularizers.** Based upon these insights, we demonstrate that deploying `TANGOS` *in tandem* with other regularizers can further improve generalization of neural networks in the tabular setting beyond that of any individual regularizer.

## 2 RELATED WORK

**Gradient Attribution Regularization.** A number of methods exist which incorporate a regularisation term to penalize the network gradients in some way. Penalizing gradient attributions is a natural approach for achieving various desirable properties in a neural network. Such methods have been in use at least since Drucker & Le Cun (1992), where the authors improve robustness by encouraging invariance to small perturbations in the input space. More recently, gradient attribution regularization has been successfully applied across a broad range of application areas. Some notable examples include encouraging the learning of robust features in auto-encoders (Rifai et al., 2011), improving stability in the training of generative adversarial networks (Gulrajani et al., 2017), and providing robustness to adversarial perturbations (Moosavi-Dezfooli et al., 2019). While many works have applied a shrinkage penalty (L2) to input gradients, Ross et al. (2017a) explore the effects of encouraging sparsity by considering an L1 penalty term. Gradient penalties may also be leveraged to compel a network to *attend* to particular human-annotated input features (Ross et al., 2017b). A related line of work considers the use of gradient aggregation methods such as Integrated Gradients (Sundararajan et al., 2017) and, typically, penalizes their deviation from a given target value (see e.g. Liu & Avci (2019) and Chen et al. (2019)). In contrast to these works, we do not require manually annotated regions upon which we constrain the network to attend. Similarly, Erion et al. (2021) provide methods for encoding domain knowledge such as smoothness between adjacent pixels in an image. We note that while these works have investigated penalizing a predictive model's output attributions, we are the first to regularize attributions on latent neuron activations. We provide an extended discussion of related works on neural network regularization more generally in Appendix A.

## 3 TANGOS

### 3.1 PROBLEM FORMULATION

We operate in the standard supervised learning setting, with $d_X$-dimensional input variables $X \in \mathcal{X} \subseteq \mathbb{R}^{d_X}$ and target output variable $Y \in \mathcal{Y} \subseteq \mathbb{R}$. Let $P_{XY}$ denote the joint distribution between input and target variables. The goal of the supervised learning algorithm is to find a predictive model, $f_\theta : \mathcal{X} \to \mathcal{Y}$ with learnable parameters $\theta \in \Theta$. The predictive model belongs to a hypothesis space $f_\theta \in \mathcal{H}$ that can map from the input space to the output space.

The predictive function is usually learned by optimizing a loss function $\mathcal{L} : \Theta \to \mathbb{R}$ using *empirical risk minimization* (ERM). The empirical risk cannot be directly minimized since the data distribution $P_{XY}$ is not known. Instead, we use a finite number of iid samples $(x, y) \sim P_{XY}$, which we refer to as the training data $\mathcal{D} = \{(x_i, y_i)\}_{i=1}^N$.

Once the predictive model is trained on $\mathcal{D}$, it should ideally predict well on out-of-sample data generated from the same distribution. However, overfitting can occur if the hypothesis space $\mathcal{H}$ is too complex and the sampling of training data does not fully represent the underlying distribution $P_{XY}$. Regularization is an approach that reduces the complexity of the hypothesis space so that more generalized functions are learned to explain the data. This leads to the following ERM:

$$\theta^* = \arg\min_{\theta \in \Theta} \frac{1}{|\mathcal{D}|} \sum_{(x,y) \in \mathcal{D}} \mathcal{L}(f_\theta(x), y) + \mathcal{R}(\theta, x, y), \tag{1}$$

that includes an additional regularization term $\mathcal{R}$ which, generally, is a function of input $x$, the label $y$, the model parameters $\theta$, and reflects prior assumptions about the model. For example, $L1$ regularization reflects the belief that sparse solutions in parameter space are more desirable.

### 3.2 NEURON ATTRIBUTIONS

Formally, attribution methods aim to uncover the importance of each input feature of a given sample to the prediction of the neural network. Recent works have demonstrated that feature attribution methods can be incorporated into the training process (Lundberg & Lee, 2017; Erion et al., 2021). These *attribution priors* optimize attributions to have desirable characteristics, including interpretability as well as smoothness and sparsity in predictions. However, these methods have exclusively investigated *output* attributions, i.e., contributions of input features to the output of a model. To the best of our knowledge, we are the first work to investigate regularization of *latent attributions*.

We rewrite our predictive function $f$ using function composition $f = l \circ g$. Here $g : \mathcal{X} \to \mathcal{H}$ maps the input to a representation $h = g(x) \in \mathcal{H}$, where $\mathcal{H} \subseteq \mathbb{R}^{d_H}$ is a $d_H$-dimensional latent space. Additionally, $l : \mathcal{H} \to \mathcal{Y}$ maps the latent representation to a label space $y = l(h) \in \mathcal{Y}$. We let $h_i = g_i(x)$, for, $i \in [d_H]$ denote the $i^{th}$ neuron in the hidden layer of interest. Additionally, we use $a_j^i(x) \in \mathbb{R}$ to denote the attribution of the $i^{th}$ neuron w.r.t. the feature $x_j$. With this notation, upper indices correspond to latent units and lower indices to features. In some cases, it will be convenient to stack all the feature attributions together in the attribution vector $a^i(x) = [a_j^i(x)]_{j=1}^{d_X} \in \mathbb{R}^{d_X}$.

Attribution methods work by using gradient signals to evaluate the contributions of the input features. In the most simplistic setting:

$$a_j^i(x) \equiv \frac{\partial h_i(x)}{\partial x_j}. \tag{2}$$

This admits a simple interpretation through a first-order Taylor expansion: if the input feature $x_j$ were to increase by some small number $\epsilon \in \mathbb{R}^+$, the neuron activation would change by $\epsilon \cdot a_j^i(x) + \mathcal{O}(\epsilon^2)$. The larger the absolute value of the gradient, the stronger the effect of a change in the input

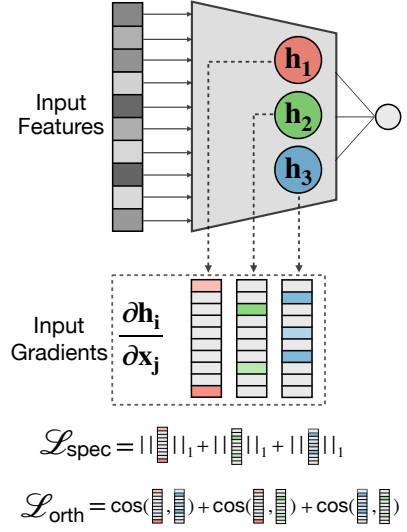

Figure 2: **Method illustration.** `TANGOS` regularizes the gradients with respect to each of the latent units.

feature. We emphasize that our method is *agnostic* to the gradient attribution method, as different methods may be more appropriate for different tasks. For a comprehensive review of different methods, assumptions, and trade-offs, see Ancona et al. (2017). For completeness, we also note another category of attribution methods is built around *perturbations*: this class of methods evaluates contributions of individual features through repeated perturbations. Generally speaking, they are more computationally inefficient due to the multiple forward passes through the neural network and are difficult to include directly in the training objective.

### 3.3 Rewarding Orthogonalization and Specialization

The main contribution of this work is proposing regularization on neuron attributions. In the most general sense, any function of any neuron attribution method could be used as a regularization term, thus encoding prior knowledge about the properties a model should have.

Specifically, the regularization term is a function of the network parameters $\theta$ and $x$, i.e., $\mathcal{R}(\theta, x)$, and encourages prior assumptions on desired behavior of the learned function. Biological sensory neurons are highly specialized. For example, certain visual neurons respond to a specific set of visual features including edges and orientations within a single receptive field. They are thus highly *selective* with *limited capacity* to react to specific physiological stimuli (Johnston & Dark, 1986; Cowan et al., 2005). Similarly, we hypothesize that neurons that are more specialized and pay attention to sparser signals should exhibit better generalization performance. We propose the following desiderata and corresponding regularization terms:

- **Specialization.** The contribution of input features to the activation of a particular neuron should be sparse, i.e., $||a^i(x)||$ is small for all $i \in [d_H]$ and $x \in \mathcal{X}$. Intuitively, in higher-dimensional settings, a few features should account for a large percentage of total attributions while others are near zero, resulting in more *specialized* neurons. We write this as a regularization term for mini-batch training:

$$\mathcal{L}_{\text{spec}}(x) = \frac{1}{B} \sum_{b=1}^{B} \frac{1}{d_H} \sum_{i=1}^{d_H} \|a^i(x_b)\|_1,$$

  where $b \in [B]$ is the batch index of $x_b \in \mathcal{X}$ and $\|\cdot\|_1$ denotes the $l_1$ norm.

- **Orthogonalization.** Different neurons should attend to non-overlapping subsets of input features given a particular input sample. To encourage this, we penalize the correlation between neuron attributions $\rho[a^i(x), a^j(x)]$ for all $i \neq j$ and $x \in \mathcal{X}$. In other words, for each particular input, we

want to discipline the latent units to attend to different aspects of the input. Then, expressing this as a regularization term for mini-batch training, we obtain:

$$\mathcal{L}_{\text{orth}}(x) = \frac{1}{B} \sum_{b=1}^{B} \frac{1}{C} \sum_{i=2}^{d_H} \sum_{j=1}^{i-1} \rho \left[ a^i(x_b), a^j(x_b) \right].$$

Here, $C$ is the number of pairwise correlations, $C = \frac{d_H \cdot (d_H - 1)}{2}$, and $\rho[a^i(x_b), a^j(x_b)] \in [0, 1]$ is calculated using the cosine similarity $\frac{|a^{i\intercal}(x_b)\, a^j(x_b)|}{||a^i(x_b)||_2 ||a^j(x_b)||_2}$ where $\|\cdot\|_2$ denotes the $l_2$ norm.

These terms can be combined into a single regularization term and incorporated into the training objective. The resulting TANGOS regularizer can be expressed as:

$$\mathcal{R}_{\text{TANGOS}}(x) = \lambda_1 \mathcal{L}_{\text{spec}}(x) + \lambda_2 \mathcal{L}_{\text{orth}}(x),$$

where $\lambda_1, \lambda_2 \in \mathbb{R}$ act as weighting terms. As this expression is computed using gradient signals, it can be efficiently implemented and minimized in any auto-grad framework.

## 4 *How* AND *Why* DOES TANGOS WORK?

To the best of our knowledge, TANGOS is the only work to explicitly regularize latent neuron attributions. A natural question to ask is (1) *How is* TANGOS *different from other regularization?* While intuitively it makes sense to enforce *specialization* of each unit and *orthogonalization* between units, we empirically investigate if other regularizers can achieve similar effects, revealing that our method regularizes a unique objective. Having established that the TANGOS objective is unique, the next question is (2) *Why does it work?* To investigate this question, we frame the set of neurons as an ensemble, and demonstrate that our regularization improves diversity among *weak learners*, resulting in improved out-of-sample generalization.

### 4.1 TANGOS REGULARIZES A UNIQUE OBJECTIVE

TANGOS encourages generalization by explicitly decorrelating and sparsifying the attributions of latent units. A reasonable question to ask is if this is unique, or if other regularizers might achieve the same objective implicitly. Two alternative regularizers that one might consider are $L2$ weight regularization and $Dropout$. Like TANGOS, weight regularization methods implicitly and partially penalize the gradients by shrinking the weights in the neural network. Additionally, $Dropout$ trains an ensemble of learners by forcing each neuron to be more independent. In Figure 3, we provide these results on the UCI temperature forecast dataset (Cho et al., 2020), in which data from 25 weather stations in South Korea is used to predict next-day peak temperature. We train a fully connected neural network for each regularization method. Specifically, we plot $\mathcal{L}_{spec}$ and $\mathcal{L}_{orth}$ for neurons in the penultimate layers and the corresponding generalization performance. We supply an extended selection of these results on additional datasets and regularizers in Appendix J.

First, we observe that TANGOS significantly decreases correlation between different neuron attributions while other regularization terms, in fact, increase them. For $L2$ weight regularization, this suggests that as the neural network weights are made smaller, the neurons increasingly attend to the same input features. A similar effect is observed for $Dropout$ - which has a logical explanation. Indeed, $Dropout$ creates redundancy by forcing each latent unit to be independent of others. Naturally, this encourages individual neurons to attend to overlapping features. In contrast, TANGOS aims to achieve specialization, such that neurons pay attention to sparse, non-overlapping features.

Additionally, we note that no alternative regularizers achieve greater attribution sparsity. This does not come as a surprise for $Dropout$, where the aim to induce redundancy in each neuron will naturally encourage individual neurons to attend to more features. While $L2$ does achieve a similar level of sparsity, this is paired with a high $\mathcal{L}_{orth}$ term indicating that, although the latent units do attend to sparse features, they appear to collapse to a solution in which they all attend to the same weighted subset of the input features. This, as we will discover in §4.2, is unlikely to be optimal for out-of-sample generalization.

Therefore, we conclude that the pairing of the specialization and orthogonality objectives in TANGOS regularizes a unique objective.

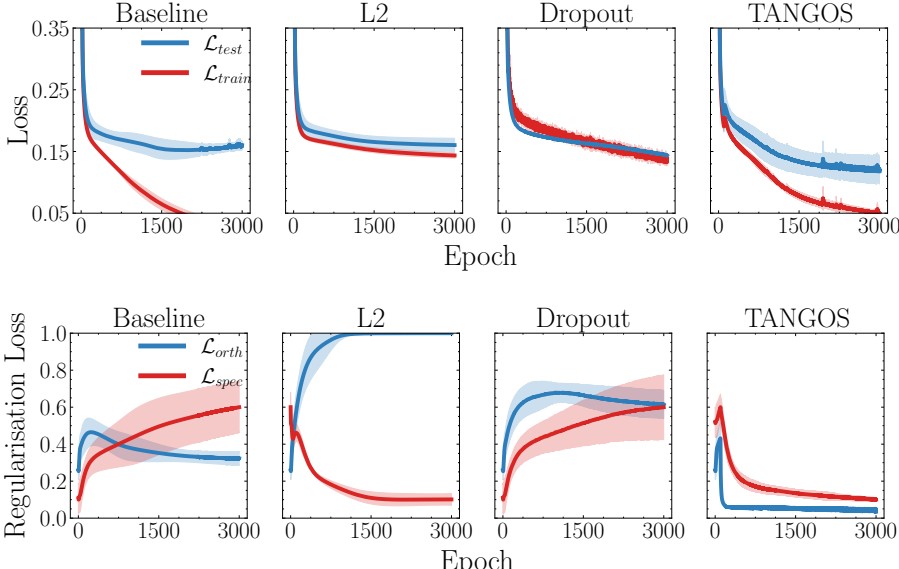

Figure 3: **Comparison of regularization objectives. (Top)** Generalization performance of key regularization techniques, **(Bottom)** corresponding neuron attributions evaluated on the test set. $L2$ and $DO$ can reduce overfitting, but neuron attributions are in fact becoming more correlated. TANGOS achieves the best generalization performance by penalizing a different objective.

## 4.2 TANGOS GENERALIZES BY INCREASING DIVERSITY AMONG LATENT UNITS

Having established how TANGOS differs from existing regularization objectives, we now turn to answer *why* it works. In this section, we provide an alternative perspective on the effect of TANGOS regularization in the context of ensemble learning. A predictive model $f(x)$ may be considered as an ensemble model if it can be written in the form $f(x) = \sum_{T_k \in \mathcal{T}} \alpha_k T_k(x)$, where $\mathcal{T}$ represents a set of basis functions sometimes referred to as *weak learners* and the $\alpha_k$'s represent their respective scalar weights. It is therefore clear that each output of a typical neural network may be considered an ensemble predictor with every latent unit in its penultimate layer acting as a weak learner in their contribution to the model's output. More formally, in this setting $T_k(x)$ is the activation of latent unit $k$ with respect to an input $x$ and $\alpha_k$ is the subsequent connection to the output activation. With this in mind, we present the following definition.

**Definition 4.1.** *Consider an ensemble regressor* $f(x) = \sum_{T_k \in \mathcal{T}} \alpha_k T_k(x)$ *trained on* $\mathcal{D} = \{(x_i, y_i)\}_{i=1}^{N}$ *where each* $(x, y)$ *is drawn randomly from* $P_{XY}$. *Additionally, the weights are constrained such that* $\sum_k \alpha_k = 1$. *Then, for a given input-label pair* $(x, y)$, *we define:*

(a) *The overall ensemble error as:* $\mathrm{Err} = (f(x) - y)^2$.

(b) *The weighted errors of the weak learners as:* $\overline{\mathrm{Err}} = \sum_k \alpha_k (T_k(x) - y)^2$.

(c) *The ensemble diversity as:* $\mathrm{Div} = \sum_k \alpha_k (T_k(x) - f(x))^2$.

Intuitively, $\overline{\mathrm{Err}}$ provides a measure of the strength of the ensemble members while Div measures the diversity of their outputs. To understand the relationship between these two terms and the overall ensemble performance, we consider Proposition 1.

**Proposition 1** (Krogh & Vedelsby 1994). *The overall ensemble error for an input-label pair* $(x, y)$ *can be decomposed into the weighted errors of the weak learners and the ensemble diversity such that:*

$$\mathrm{Err} = \overline{\mathrm{Err}} - \mathrm{Div}. \tag{3}$$

This decomposition provides a fundamental insight into the success of ensemble methods: an ensemble's overall error is reduced by decreasing the average error of the individual weak learners and increasing the diversity of their outputs. Successful ensemble methods explicitly increase ensemble diversity when training weak learners by, for example, sub-sampling input features (random forest, Breiman, 2001), sub-sampling from the training data (bagging, Breiman, 1996) or error-weighted input importance (boosting, Bühlmann, 2012).

Returning to the specific case of neural networks, it is clear that TANGOS provides a similar mechanism of increasing diversity among the latent units that act as weak learners in the penultimate

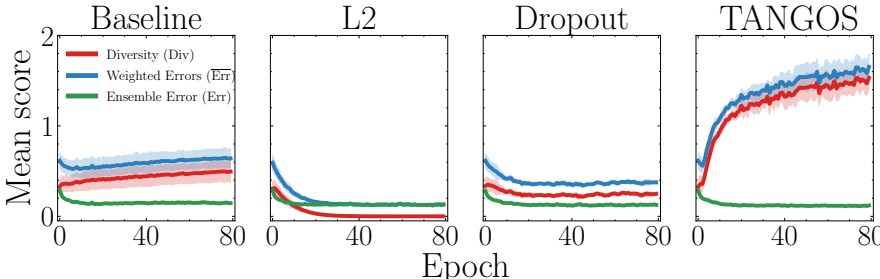

Figure 4: **Neuron Diversity.** Overall ensemble error and decomposition in terms of diversity and average error of the weak learners. Note while all methods achieve low overall error, TANGOS is the only method that does so by increasing the diversity among the latent units.

layer. By forcing the latent units to attend to sparse, uncorrelated selections of features, the learned ensemble is encouraged to produce diverse learners whilst maintaining coverage of the entire input space in aggregate. In Figure 4, we demonstrate this phenomenon in practice by returning to the UCI temperature forecast regression task. We provide extended results in Appendix J. We train a fully connected neural network with two hidden layers with the output layer weights constrained such that they sum to 1. We observe that regularizing with TANGOS increases diversity of the latent activations resulting in improved out-of-sample generalization. This is in contrast to other typical regularization approaches which also improve model performance, but exclusively by attempting to reduce the error of the individual ensemble members. This provides additional motivation for applying TANGOS in the tabular domain, an area where traditional ensemble methods have performed particularly well.

Table 1: **Stand-Alone Regularization.** Comparison of regularizers on regression and classification in terms of test MSE and NLL. All models are trained on real-world datasets using 5-fold cross-validation and final evaluation reported on a held-out test set. **Bold** indicates the best performance. The average rank of each method across both regression and classification is included in the final row of the respective tables.

| Dataset | Baseline | L1 | L2 | DO | BN | IN | MU | TANGOS |
|---------|----------|-----|-----|-----|-----|-----|-----|--------|
| Regression (Mean Squared Error) | | | | | | | | |
| FB | 0.037 | 0.081 | **0.029** | 0.060 | 0.699 | 0.043 | 0.147 | 0.032 |
| BH | 0.192 | 0.197 | 0.183 | 0.209 | 0.190 | 0.215 | 0.286 | **0.166** |
| WE | 0.118 | 0.096 | 0.099 | 0.097 | **0.090** | 0.101 | 0.146 | 0.093 |
| BC | 0.323 | 0.263 | 0.277 | 0.282 | 0.294 | 0.308 | 0.323 | **0.244** |
| WQ | 0.673 | 0.641 | 0.644 | 0.658 | 0.639 | 0.669 | 0.713 | **0.637** |
| SC | 0.422 | 0.408 | 0.411 | 0.423 | 0.410 | 0.434 | 0.547 | **0.387** |
| FF | 1.274 | 1.280 | 1.274 | 1.266 | 1.330 | **1.201** | 1.289 | 1.276 |
| PR | 0.624 | 0.611 | 0.580 | 0.592 | 0.647 | 0.591 | 0.745 | **0.573** |
| ST | 0.419 | 0.416 | 0.418 | 0.387 | 0.461 | 0.539 | **0.380** | 0.382 |
| AB | 0.345 | 0.319 | 0.332 | **0.312** | 0.348 | 0.355 | 0.366 | 0.325 |
| Avg Rank | 5.4 | 3.8 | 3.4 | 4.0 | 5.0 | 5.5 | 7.1 | **1.9** |
| Classification (Mean Negative Log-likelihood) | | | | | | | | |
| HE | 0.490 | 0.472 | 0.431 | 0.428 | 0.459 | 0.435 | **0.416** | 0.426 |
| BR | 0.074 | 0.070 | 0.070 | 0.078 | 0.080 | 0.071 | 0.095 | **0.069** |
| CE | 0.519 | **0.395** | 0.407 | 0.436 | 0.604 | 0.457 | 0.472 | 0.408 |
| CR | 0.464 | 0.405 | 0.402 | 0.456 | 0.460 | 0.481 | 0.448 | **0.369** |
| HC | 0.320 | 0.222 | 0.226 | 0.237 | 0.257 | 0.312 | 0.248 | **0.215** |
| AU | 0.448 | 0.442 | 0.385 | 0.405 | 0.549 | 0.479 | 0.478 | **0.379** |
| TU | 1.649 | 1.633 | 1.613 | 1.621 | **1.484** | 1.646 | 1.657 | 1.495 |
| EN | 1.040 | 1.040 | 1.042 | 1.058 | 1.098 | 1.072 | 1.065 | **0.974** |
| TH | 0.700 | 0.506 | **0.500** | 0.714 | 0.785 | 0.638 | 0.618 | 0.513 |
| SO | 0.606 | **0.238** | 0.382 | 0.567 | 0.484 | 0.540 | 0.412 | 0.371 |
| Avg Rank | 6.4 | 3.0 | 2.7 | 4.8 | 6.3 | 6.0 | 5.2 | **1.7** |

## 5 EXPERIMENTS

In this section, we empirically evaluate TANGOS as a regularization method for improving generalization performance. We present our benchmark methods and training architecture, followed by extensive results on real-world datasets. There are a few main aspects that deserve empirical

investigation, which we investigate in turn: ▶ **Stand-alone performance.** §5.1 Comparing the performance of `TANGOS`, where the focus is on applying it as a stand-alone regularizer, to a variety of benchmarks on a suite of real-world datasets. ▶ **In tandem performance.** §5.2 Motivated by our unique regularization objective and our analysis in §4, we demonstrate that applying `TANGOS` *in conjunction* with other regularizers can lead to even greater gains in generalization performance. ▶ **Modern architectures.** §5.3 We evaluate performance on a state-of-the-art tabular architecture and compare to boosting. All experiments were run on NVIDIA RTX A4000 GPUs. Code is provided on Github[1][2].

**TANGOS.** We train `TANGOS` regularized models as described in Algorithm 1 in Appendix F. For the specialization parameter we search for $\lambda_1 \in \{1, 10, 100\}$ and for the orthogonalization parameter we search for $\lambda_2 \in \{0.1, 1\}$. For computational efficiency, we apply a sub-sampling scheme where 50 neuron pairs are randomly sampled for each input (for further details see Appendix F).

**Benchmarks.** We evaluate `TANGOS` against a selection of popular regularizer benchmarks. First, we consider weight decay methods **L1** and **L2** regularization, which sparsify and shrink the learnable parameters. For the regularizers coefficients, we search for $\lambda \in \{0.1, 0.01, 0.001\}$ where regularization is applied to all layers. Next, we consider Dropout (**DO**), with drop rate $p \in \{10\%, 25\%, 50\%\}$, and apply DO after every dense layer during training. We also consider implicit regularization in batch normalization (**BN**). Lastly, we evaluate data augmentation techniques Input Noise (**IN**), where we use additive Gaussian noise with mean 0 and standard deviation $\sigma \in \{0.1, 0.05, 0.01\}$ and MixUp (**MU**). Furthermore, each training run applies early stopping with patience of 30 epochs. In all experiments, we use 5-fold cross-validation to train and validate each benchmark. We select the model which achieves the lowest validation error and provide a final evaluation on a held-out test set.

## 5.1 GENERALIZATION: STAND-ALONE REGULARIZATION

For the first set of experiments, we are interested in investigating the individual regularization effect of `TANGOS`. To ensure a fair comparison, we evaluate the generalization performance on held-out test sets across a variety of datasets.

**Datasets.** We employ 20 real-world tabular datasets from the UCI machine learning repository. Each dataset is split into $80\%$ for cross-validation and the remaining $20\%$ for testing. The splits are standardized on just the training data, such that features have mean 0 and standard deviation 1 and categorical variables are one-hot encoded. See Appendix L for further details on the 20 datasets used.

**Training and Evaluation.** To ensure a fair comparison, all regularizers are applied to an MLP with two ReLU-activated hidden layers, where each hidden layer has $d_H + 1$ neurons. The models are trained using Adam optimizer with a dataset-dependent learning rate from $\{0.01, 0.001, 0.0001\}$ and are trained for up to a maximum of 200 epochs. For regression tasks, we report the average Mean Square Error (MSE) and, on classification tasks, we report the average negative log-likelihood (NLL).

**Results.** Table 1 provides the benchmarking results for individual regularizers. We observe that `TANGOS` achieves the best performance on $10/20$ of the datasets. We also observe that on 6 of the remaining datasets, `TANGOS` ranks second. This is also illustrated by the ranking plot in Appendix H. There we also provide a table displaying standard errors. As several results have overlapping error intervals, we also assess the magnitude of improvement by performing a non-parametric Wilcoxon signed-rank sum test (Wilcoxon, 1992) paired at the dataset level. We compare `TANGOS` to the best-performing baseline method (L2) as a one-tailed test for both the regression and classification results obtaining p-values of 0.006 and 0.026 respectively. This can be interpreted as strong evidence to suggest the difference is statistically significant in both cases. Note that a single regularizer is seldom used by itself. In addition to a stand-alone method, it remains to be shown that `TANGOS` brings value when used with other regularization methods. This is explored in the next section.

## 5.2 GENERALISATION: IN TANDEM REGULARIZATION

Motivated by the insights described in §4, a natural next question is whether `TANGOS` can be applied in conjunction with existing regularization to unlock even greater generalization performance. In this set of experiments, we investigate this question.

---

[1]`https://github.com/alanjeffares/TANGOS`
[2]`https://github.com/vanderschaarlab/TANGOS`

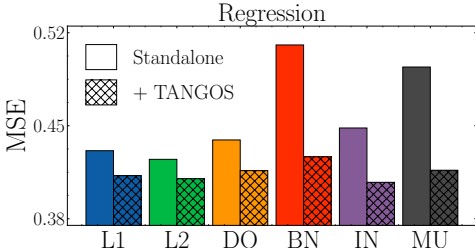 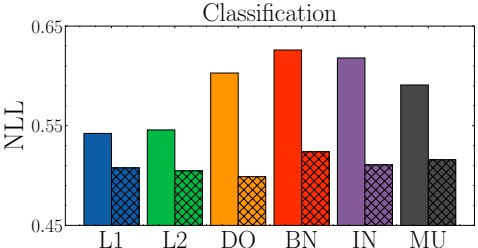

Figure 5: **In Tandem Regularization.** Aggregated errors across the 10 regression datasets (left) and the 10 classification datasets (right). In all cases, the addition of TANGOS provides superior performance over the standalone regularizer.

**Setup.** The setting for this experiment is identical to §5.1 except now we consider the six baseline regularizers *in tandem* with TANGOS. We examine if pairing our proposed regularizer with existing methods results in even greater generalization performance. We again run 5-fold cross-validation, searching over the same hyperparameters, with the final models evaluated on a held-out test set.

**Results.** We summarize the aggregated results over the datasets for each of the six baseline regularizers in combination with TANGOS in Figure 5. Consistently across all regularizers in both the regression and the classification settings, we observe that adding TANGOS regularization improves test performance. We provide the full table of results in the supplementary material. We also note an apparent interaction effect between certain regularizers (i.e. input noise for regression and dropout for classification), where methods that seemed to not be particularly effective as stand-alone regularizers become the best-performing method when evaluated in tandem. The relationship between such regularizers provides an interesting direction for future work.

## 5.3 CLOSING THE GAP ON BOOSTING

In this experiment, we apply TANGOS regularization to a state-of-the-art deep learning architecture for tabular data (Gorishniy et al., 2021) and evaluate its contribution towards producing competitive performance against leading boosting methods. We provide an extended description of this experiment in Appendix B and results in Table 2. We find that TANGOS provides moderate gains in this setting, improving performance relative to state-of-the-art boosting methods. Although boosting approaches still match or outperform deep learning in this setting, in Appendix C we argue that deep learning may also be worth pursuing in the tabular modality for its other distinct advantages.

Table 2: **FT-Transformer Architecture and Boosting.** Adding TANGOS regularization can contribute to closing the gap between state-of-the-art tabular architectures and leading boosting methods. We report mean accuracy ± standard deviation.

| Setting | Dataset | FT-Transformer | | Boosting | |
|---|---|---|---|---|---|
| | | Baseline | + TANGOS | XGBoost | CatBoost |
| Default | Jannis | $0.714 \pm 0.002$ | $\mathbf{0.720} \pm 0.000$ | $0.711 \pm 0.000$ | $\mathbf{0.724} \pm 0.001$ |
| | Higgs | $0.721 \pm 0.002$ | $\mathbf{0.723} \pm 0.000$ | $0.717 \pm 0.000$ | $\mathbf{0.728} \pm 0.001$ |
| Tuned | Jannis | $0.720 \pm 0.001$ | $\mathbf{0.727} \pm 0.001$ | $0.724 \pm 0.000$ | $\mathbf{0.727} \pm 0.001$ |
| | Higgs | $0.727 \pm 0.002$ | $\mathbf{0.729} \pm 0.002$ | $0.728 \pm 0.001$ | $\mathbf{0.729} \pm 0.002$ |

## 6 DISCUSSION

In this work, we have introduced TANGOS, a novel regularization method that promotes specialization and orthogonalization among the gradient attributions of the latent units of a neural network. We showed *how* this regularization objective is distinct from other popular methods and motivated *why* it provides out-of-sample generalization. We empirically demonstrated TANGOS utility with extensive experiments. This work raises several exciting avenues for **future work** including (1) developing TANGOS beyond the tabular setting (e.g. images), (2) investigating alternative efficient methods for achieving specialization and orthogonalization, (3) proposing other latent gradient attribution regularizers, (4) augmenting TANGOS for specific applications such as multi-modal learning or increased interpretability (see Appendix C).

## ACKNOWLEDGMENTS

We thank the anonymous ICLR reviewers as well as members of the van der Schaar lab for many insightful comments and suggestions. Alan Jeffares is funded by the Cystic Fibrosis Trust. Tennison Liu would like to thank AstraZeneca for their sponsorship and support. Fergus Imrie and Mihaela van der Schaar are supported by the National Science Foundation (NSF, grant number 1722516). Mihaela van der Schaar is additionally supported by the Office of Naval Research (ONR).

## REPRODUCIBILITY STATEMENT

We have attempted to make our experimental results easily reproducible by both a detailed description of our experimental procedure and providing the code used to produce our results (`https://github.com/alanjeffares/TANGOS`). Experiments are described in Section 5 with further details in Appendices F and L. All datasets used in this work can be freely downloaded from the UCI repository (Dua et al., 2017) with specific details provided in Appendix L.

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

## A    EXTENDED RELATED WORKS

**Neural Network Regularization.** Regularization methods seek to penalize complexity and impose a form of smoothness on a model. This may be cast as expressing a prior belief over the hypothesis space of a neural network which attempts to aid generalization. ▶ **Categories.** A vast array of regularization methods have been proposed throughout the literature (for a comprehensive taxonomy see e.g. Kukačka et al., 2017). Modern nomenclature typically includes broad modeling decisions such as choice of architecture, loss function, and optimization method under the umbrella of regularization. Additionally, many regularization techniques augment the training data using methods such as input noise (Krizhevsky et al., 2012) or MixUp (Zhang et al., 2018). Dropout (Hinton et al., 2012) and related approaches that augment a hidden representation of the input may also be included in this category. Possibly a more conventional category of regularization is that which adds explicit penalty term(s) to the loss function. These terms might penalize the network weights directly to shrink or sparsify their values as in L2 (Hoerl & Kennard, 1970) and L1 (Tibshirani, 1996), respectively. Alternatively, network outputs may be penalized to, for example, reduce overconfidence (Pereyra et al., 2017). ▶ **Weight Orthogonalization.** A number of works have studied the orthogonalization of network weights via various weight penalization methods (Bansal et al., 2018). More recent work in Liu et al. (2021) proposed to learn an orthogonal transformation of the randomly initialized incoming weights to a given neuron. In contrast, this work seeks to ensure that the gradients of different latent neurons with respect to a given input vector are orthogonal. ▶ **Combination.** Compositions of multiple regularization methods are extensively applied in practice. An early example in the regression setting is the elastic net penalty (Zou & Hastie, 2005) which attempts to combine sparsity with shrinkage in the coefficients. More recent work has demonstrated the effectiveness of combining several regularization terms on tabular data (Kadra et al., 2021), a domain in which neural networks superiority had previously been less convincing.

## B    TABULAR ARCHITECTURES AND BOOSTING

While non-neural methods such as XGBoost (Chen & Guestrin, 2016) and CatBoost (Prokhorenkova et al., 2018) are still considered state of the art for tabular data (Grinsztajn et al., 2022), much progress has been made in recent years to close the gap. Furthermore, differing learning paradigms have various strengths and weaknesses outside of maximum generalization performance, which is often a consideration in practical applications. While boosting methods boast excellent computational efficiency and strong out-of-the-box performance, neural networks have unique utility in, for example, multi-modal learning (Ramachandram & Taylor, 2017), meta-learning (Hospedales et al., 2021) and certain interpretability methods (Zhang et al., 2021). In this section, we provide additional experiments applying TANGOS to a state-of-the-art transformer architecture for tabular data proposed in Gorishniy et al. (2021). Specifically, this architecture combines a Feature Tokenizer which transforms features into embeddings with a multi-layer Transformer (Vaswani et al., 2017). We compare this FT-Transformer architecture to boosting methods in the *default* setting where we evaluate out-of-the-box performance and the *tuned* setting where we jointly optimize the Transformer along with its baseline regularizers. We describe these two settings in more detail next.

**Default Setting.** In this setting, we use a 3-layer Transformer with a 32-dimensional feature embedding size and 4 attention heads. Following the original paper we use Reglu activations, a hidden layer size of 43 corresponding to a ratio of $\frac{4}{3}$ with the embedding size, Kaiming initialization (He et al., 2015), and AdamW optimizer (Loshchilov & Hutter, 2017). Finally, we apply a learning rate of 0.001. We compare this architecture with and without TANGOS regularization applied which we refer to as "Baseline" and "+ TANGOS" respectively. We set $\lambda_1 = 1$ and $\lambda_2 = 0.01$ which were found to be reasonable default values for specialization and orthogonalization in our experiments in Section 5.

**Tuned Setting.** Here we apply ten iterations of random search tuning over the same hyperparameters as in the original work with those achieving the best validation performance selected. We then evaluate this combination by training over three seeds and perform their final evaluations on a held-out test set. We search using the same distributions as in the original work and consider the following ranges. L2 regularization $\in [1e-06, 1e-03]$, residual dropout $\in [0.0, 0.2]$, hidden layer dropout $\in [0.0, 0.5]$, attention dropout $\in [0.0, 0.5]$, hidden layer to feature embedding dimension ratio $\in [1.0, 3.0]$, embedding dimension $\in [16, 48]$, number of layers $\in [1, 3]$, learning rate $\in [1e-04, 1e-03]$. In

the "+ TANGOS" setting we also include $\lambda_1 \in [0.001, 10]$ and $\lambda_2 \in [0.0001, 1]$ with a log uniform distribution. All remaining architecture choices are consistent with the default setting and the original work.

We ran our experiments on the Jannis (Guyon et al., 2019) and Higgs (Baldi et al., 2014) datasets. These are both classification datasets consisting of 83733 and 98050 examples respectively. These datasets were selected as they represent a significant number of input examples along with a middling number of input features relative to the other tabular datasets explored in this work (54 and 28 respectively). We follow the experimental protocol of the boosting comparison in Grinsztajn et al. (2022) using the same training, validation, and test splits and reporting mean test accuracy over three runs. Therefore we obtain the same results for boosting as reported in that work.

The results of this experiment are reported in Table 2 where we find that TANGOS does indeed have a positive effect on the FT-Transformer performance although, consistent with the original work, we found that regularization only provides modest gains at best with this architecture. While we do not claim that TANGOS regularization results in neural networks that outperform Boosting methods, these results indicate that TANGOS regularization can contribute to closing the gap and may play a key role when combined with other methods as highlighted in Kadra et al. (2021). We believe this to be an important area for future research and, in particular, expect that architecture-specific developments of the ideas presented in this work may provide further improvements on the results obtained in this section.

## C  Motivation for Deep Learning on Tabular Data

Several works have argued that boosting methods generally achieve superior performance to even state-of-the-art deep learning architectures for tabular data (Grinsztajn et al., 2022; Shwartz-Ziv & Armon, 2022). However, this is in contrast to recent findings for transformer style architectures in Gorishniy et al. (2021), especially with appropriate feature embeddings (Gorishniy et al., 2022) and sufficient pretraining (Rubachev et al., 2022). We defer from this discussion to highlight a selection of reasons to consider deep learning methods for tabular data *beyond* straightforward improvements in predictive performance. In particular, we include a number of deep learning paradigms that are difficult to analogize for non-neural models and have been successfully applied to tabular data.

**Multi-modal learning** refers to the task of modeling data inputs that consist of multiple data modalities (e.g. image, text, tabular). As one might intuit, jointly modeling these multiple modalities can result in better performance than independently predicting from each of them (Ramachandram & Taylor, 2017; Guo et al., 2019). Deep learning provides a uniquely natural method of combining modalities with the advantages of (1) modality-specific encoders, (2) that are fused into a joint downstream representation and trained end-to-end with backpropagation, and (3) superior modeling performance in many modalities such as images and natural language. Healthcare is a domain in which multi-modal learning is particularly salient (Acosta et al., 2022). Recent work in Wu et al. (2022) showed that jointly modeling tabular clinical records using an MLP together with medical images using a CNN outperforms the non-multi-modal baselines. Elsewhere in Tang et al. (2020), a multi-modal approach is taken in combining input modalities based on the preprocessing of functional magnetic resonance imaging and region of interest time series data for the diagnosis of autism spectrum disorder. A resnet-18 encodes one modality while an MLP encodes the other, resulting in superior performance when analyzed in an ablation study. In this setting, progress in modeling each of the individual modalities is likely to result in better performance of the system as a whole. Interestingly, Ramachandram & Taylor (2017) identified regularization techniques for improved cross-modality learning as an important research direction. We believe that further development of the ideas presented in this work could provide a powerful tool for balancing how models attend to multiple input modalities.

**Meta-learning** aims to distill the experience of multiple learning episodes across a distribution of related tasks to improve learning performance on future tasks (Hospedales et al., 2021). Deep learning-based approaches have seen great success as a solution to this problem in a variety of fields. In the tabular domain, with careful consideration of the shared information between tasks, recent works have also shown promising results in this direction by developing methods for transferring deep tabular models across tables (Wang & Sun, 2022; Levin et al., 2022). In particular, in Levin et al. (2022) it was noted that "representation learning with deep tabular models provides significant

gains over strong GBDT baselines", also finding that "the gains are especially pronounced in low data regimes".

**Interpretability** is an important area of deep learning research aiming to provide users with the ability to understand and reason about model outputs. Certain classes of interpretability methods have recently been developed that provide distinct forms of interpretability relying on the hidden representations of neural networks. In such models, probing the representation space of a deep model permits a new type of interpretation. For instance, Kim et al. (2018) studies how human concepts are represented by deep classifiers. This makes it possible to analyze how the classes predicted by the model relate to human understandable concepts. For example, one can verify if the stripe concept is relevant for a CNN classifier to identify a zebra, as demonstrated in the paper. Another example is Crabbé et al. (2021), which proposes to explain a given example with reference to a freely selected set of other examples (potentially from the same dataset). A user study was carried out in this work which concluded that, among non-technical users, this method of explanation does affect their confidence in the model's prediction. These powerful methods crucially rely on the model's representation space, which effectively assumes that the model is a deep neural network.

**Representation learning** more generally provides access to several other methods from deep learning to the tabular domain. A number of works have used deep learning approaches to map inputs to embeddings which can be useful for downstream applications. SuperTML (Sun et al., 2019) and Zhu et al. (2021) map tabular inputs to image-like embeddings that can therefore be passed to image architectures such as CNNs. Other self-supervised methods include VIME (Yoon et al., 2020) which applies input reconstruction, SubTab (Ucar et al., 2021) which suggests a multi-view reconstruction task and SCARF (Bahri et al., 2021) which takes a contrastive approach. Representation learning approaches such as these have proven successful on downstream tabular data tasks such as uncertainty quantification (Seedat et al., 2023), federated learning (He et al., 2022), anomaly detection (Liang et al., 2022), and feature selection (Lee et al., 2022).

## D  TANGOS BEHAVIOR ANALYSIS

In this section, we apply TANGOS to a simple image classification task using a convolutional neural network (CNN) and provide a qualitative analysis of the behavior of the learned network. This analysis is conducted on the MNIST dataset (LeCun et al., 1998) using the recommended split resulting in 60,000 training and 10,000 validation examples.

In this experiment, we train a standard CNN architecture (as described in Table 3) with a penultimate hidden layer of 10 neurons for 10 epochs with Adam optimizer and a learning rate of 0.001. We also apply L2 regularization with weight 0.001. After each epoch, the model is evaluated on the validation set where the epoch achieving the best validation performance is stored for further analysis. Two models are trained under this protocol. One model which applied TANGOS to the penultimate hidden layer with $\lambda_1 = 100$, $\lambda_2 = 0.1$ and $M = 25$ and a baseline model which does not apply TANGOS.

Table 3: MNIST Convolutional Neural Network Architecture.

| Layer Type | Hyperparameters | Activation Function |
|---|---|---|
| Conv2d | Input Channels:1 ; Output Channels:16 ; Kernel Size:5 ; Stride:2 ; Padding:1 | ReLU |
| Conv2d | Input Channels:16 ; Output Channels:32 ; Kernel Size:5 ; Stride:2 ; Padding:1 | ReLU |
| Flatten | Start Dimension:1 | |
| Linear | Input Dimension: 512 ; Output Dimension: 256 | ReLU |
| Linear | Input Dimension: 256 ; Output Dimension: 10 | |
| Linear | Input Dimension: 10 ; Output Dimension: 10 | |

In this section, we examine the gradients of each of the 10 neurons in the penultimate hidden layer with respect to each of the input dimensions of a given image. TANGOS is designed to reward orthogonalization and specialization of these gradient attributions which can be evaluated qualitatively by inspection. In all plots that follow we apply a min-max scaling across all hidden units for a fair comparison. Both strong positive and strong negative values for attributions may be interpreted as

a latent unit attending to a given input dimension. In Figure 6 we provide results for the baseline model applied to a test image where, in line with similar analyses in previous works such as Crabbé & van der Schaar (2022), we note that the way in which hidden units attend to the input is highly entangled. In contrast to this, in Figure 7, we include the same plot for the TANGOS trained model on the same image. In this case, each hidden unit does indeed produce relatively sparse and orthogonal attributions as desired. These results were consistent across the test images.

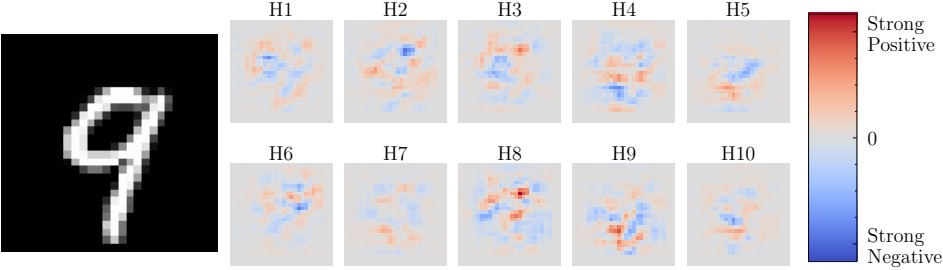

Figure 6: **Without TANGOS Training.** Gradient attributions with respect to each of the 10 hidden neurons. These results suggest significant overlap among the gradient attributions.

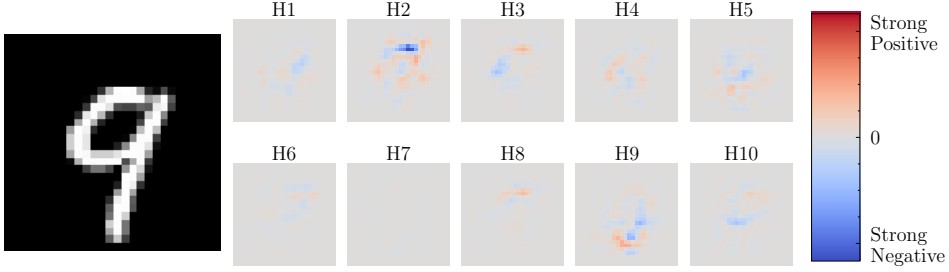

Figure 7: **With TANGOS Training.** TANGOS encourages gradient attributions to be sparse with minimal overlap.

We can glean further insight into the TANGOS trained model by examining the role of individual neurons across multiple test images. In Figure 8, we provide the gradient attributions for hidden neuron 5 (H5) from our previous discussion across twelve test images. This neuron appears to discriminate between an open or a closed loop at the lower left of the digit. Indeed this is a key aspect of distinction between the set of digits $\{2, 6, 8, 0\}$ (first row) and $\{9, 5, 3\}$ (second row). We also include digits where this visual feature is less useful as they contain no lower-left loop either open or closed (third row). This hypothesis can be further examined by analyzing the values of these activations. We note that the first two rows typically have higher magnitude with opposite signs while the third row has lower magnitude activations. In Table 4 we summarize the effect of these activation scores on class probabilities by accounting for the weights connecting to each of the ten classes. As one might expect, the weight connections between the hidden neuron and classes on the first row and the second row have opposite signs indicating that neuron 5 does indeed discriminate between these classes.

## E    PERFORMANCE WITH INCREASING DATA SIZE

In this section, we evaluate TANGOS performance with an increasing number of input examples. To do this we use the Dionis dataset, which was the largest benchmark dataset proposed in Kadra et al. (2021) with 416,188 examples. As in that work, we set aside 20% for testing with the remaining data further split into 80% training and 20% validation. The data was standardized to have zero mean and unit variance with statistics calculated on the training data. We then consider using various proportions (10%, 50%, 100%) of the training data to train an MLP with and without TANGOS regularization. We also evaluate the best-performing regularization method, L2, from our experiments in Section 5. For both regularization methods, we train three hyperparameter settings at each proportion and evaluate the best performing of the three on the test set. For TANGOS we consider $\{(\lambda_1 = 1, \lambda_2 = 0.01), (\lambda_1 = 1, \lambda_2 = 0.1), (\lambda_1 = 10, \lambda_2 = 0.1)\}$ and for L2 we consider

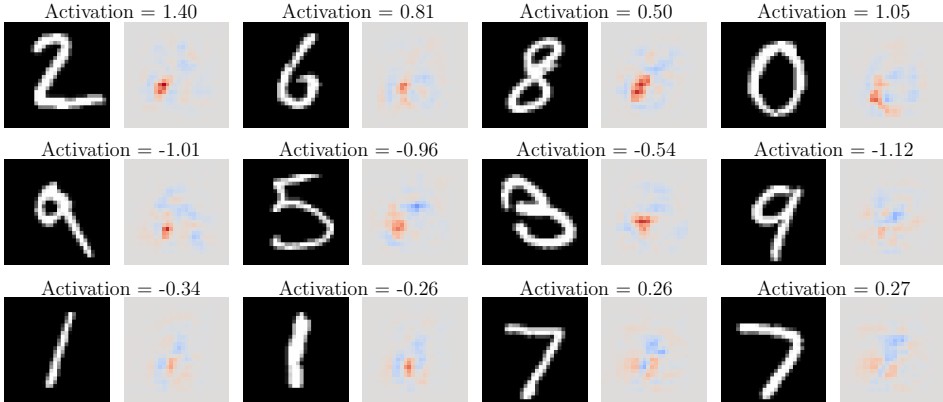

Figure 8: **Hidden Neuron 5.** This neuron attempts to discriminate whether inputs contain a closed loop on the lower left of their digit. Inputs with a closed lower loop incur highly positive activations (first row). Inputs with open lower loops incur highly negative activations (second row). While ambiguous inputs with no lower loop at all tend to produce low-magnitude activations (third row).

Table 4: **Neuron 5 Class Weights.** Weights connecting neuron 5 to each of the ten classes and a summary of their combined effect with the neuron activation. The classification influence column provides a categorical indication of the magnitude of the contribution to each class output for a fixed activation magnitude. This is determined by the magnitude of the connecting weight where: Low $\in [0, 0.59]$, Medium $\in [0.59, 1.17]$, and High $\in [1.17, 1.76]$.

| Class label | Weight connection | Increases class probability if activation is | Classification influence |
|---|---|---|---|
| 0 | 1.0883 | Positive | Medium |
| 1 | -0.6826 | Negative | Medium |
| 2 | 1.5298 | Positive | High |
| 3 | -1.5862 | Negative | High |
| 4 | -0.2516 | Negative | Low |
| 5 | -0.6008 | Negative | Medium |
| 6 | 0.9065 | Positive | Medium |
| 7 | 0.3524 | Positive | Low |
| 8 | 0.7362 | Positive | Medium |
| 9 | -1.7608 | Negative | High |

$\lambda \in \{0.01, 0.001, 0.0001\}$. We repeat this procedure for 6 runs and report the mean test accuracy. The MLP contained three ReLU-activated hidden layers of 400, 100, and 10 hidden units, respectively.

We include the results of this experiment in Figure 9. Consistent with our experiments in Section 5, we find that TANGOS outperforms both the baseline model and the strongest baseline regularization method across all proportions of the data. These results are indicative that TANGOS remains similarly effective across both small and large datasets in the tabular domain.

# F  APPROXIMATION AND ALGORITHM

Calculating the attribution of the latent units with respect to the input involves computing the Jacobian matrix, which can be computed in $\mathcal{O}(1)$ time and has memory complexity $\mathcal{O}(d_H d_X)$. The computational complexity of calculating $\mathcal{L}_{orth}$ is $\mathcal{O}(d_H^2)$ (i.e. all pairwise computation between latent units). While the calculation can be efficiently parallelized, this still becomes impractically expensive with higher dimensional layers. To address this, we introduce a relaxation by randomly subsampling pairs of neurons to calculate attribution similarity. We denote by $I$ denote the set of all possible pairs of neuron indices, $I = \{(i, j) \, \forall \, i, j \in [d_H] \text{ and } i \neq j\}$. Further, we let $M$ denote a randomly sampled subset of $I$, $M \subseteq I$. We devise an approximation to the regularization term, denoted by $\mathcal{L}'_{orth}$, by estimating the penalty on the subset $M$, where the size of $M$ can be chosen to

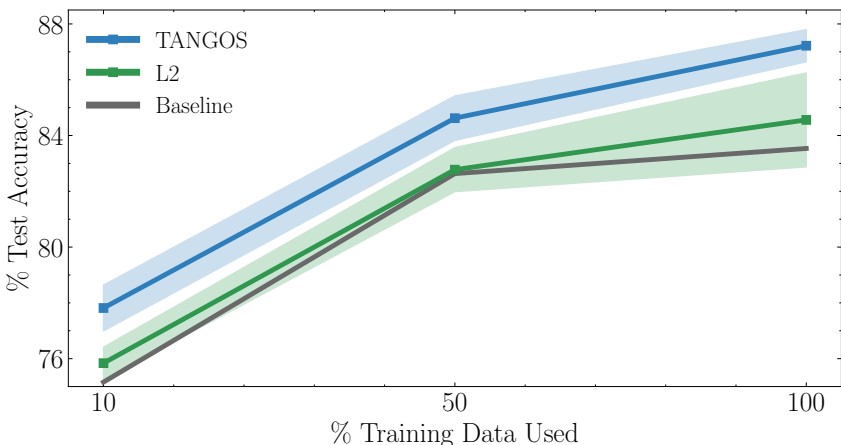

Figure 9: **Performance Gains With Increasing Data Size.** Training with various proportions of training data from the 416,188 examples of the Dionis dataset, we find the relative boost in performance from `TANGOS` to be consistent.

balance computational burden with more faithful estimation:

$$\mathcal{L}'_{orth}(x) = \frac{1}{B} \sum_{b=1}^{B} \frac{1}{|M|} \sum_{(i,j) \in M} \rho[a^i(x_b), a^j(x_b)]$$

This reduces the complexity of calculating $\mathcal{L}_{orth}$ from $\mathcal{O}(d_H^2)$ to $\mathcal{O}(M)$. For our experimental results described in Tables 1, 6 and 7, we use $|M| = 50$. We empirically demonstrate that this approximation still leads to strong results in real-world experiments. The overall training procedure is described in Algorithm 1.

---

**Algorithm 1** `TANGOS` regularization
___

    **Result:** Learned parameters $\theta$
    **Input:** $\lambda_1, \lambda_2,$ training data $\mathcal{D}$, learning rate $\eta$;
    Initialise $\theta$;
    **while** not converged **do**
        Sample $\mathcal{D}_{mini}$ from $\mathcal{D}$;
        $\hat{\mathcal{L}}(f_\theta(x), y) = \mathbb{E}_{(x,y) \sim \mathcal{D}_{mini}}[\mathcal{L}(f_\theta(x), y)]$;
        $\hat{\mathcal{R}}(x) = \lambda_1 \mathbb{E}_{x \sim \mathcal{D}_{mini}}[\mathcal{L}_{spec}(x)] + \lambda_2 \mathbb{E}_{x \sim \mathcal{D}_{mini}}[\mathcal{L}'_{orth}(x)]$;
        $\theta \leftarrow \theta + \eta \nabla_\theta \left[ \hat{\mathcal{L}}(f_\theta(x), y) + \hat{\mathcal{R}}(x) \right]$;
    **end while**

---

Additionally, we provide an empirical analysis of `TANGOS` designed to evaluate the effectiveness of our proposed subsampling approximation with respect to generalization performance and computational efficiency as the number of sampled neuron pairs $M$ grows. Furthermore, we analyze the computational efficiency of our method as the number of latent units grows, evaluating the method's capacity to scale to large models.

All experiments are run on the BC dataset which we split into 80% training and 20% validation. We fix $\lambda_1 = 100$ and $\lambda_2 = 0.1$ throughout our experiments. We run each experiment over 10 random seeds and report the mean and standard deviation. All remaining experimental details are consistent with our experiments in Section 5. We note that our implementation of `TANGOS` is not optimized to the same extent as the Pytorch (Paszke et al., 2019) implementation of L2 to which we compare, and therefore we may consider our relative computational performance to be a loose upper bound on a truly optimized version.

In Figure 10 (left), we report the relative increase in compute time per epoch as we increase the number of sampled pairs. As theory would suggest, this growth is linear. A natural follow-up question

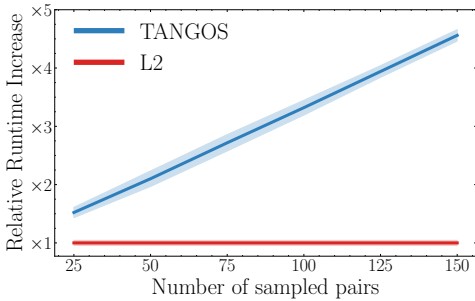 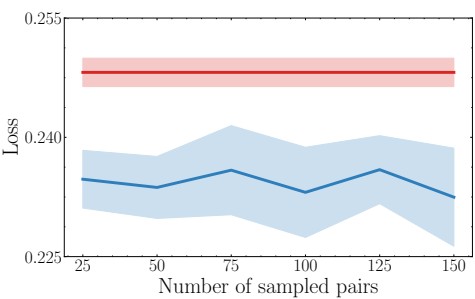

Figure 10: **Sampling efficiency.** Runtime increases linearly with the number of sampled pairs (left) while better generalization performance is maintained even for low sampling rates (right). The benefits of TANGOS can be realized using our proposed sampling approximation with comparable runtime to even the most efficient existing regularization approaches.

is the extent to which model performance is affected by decreasing the number of sampled pairs. In Figure 10 (right), we observe that even very low sampling rates still result in excellent performance. Based on these results, our recommendation for practitioners is that while increasing the sampling rate can lead to marginal improvements in performance, relatively low sampling rates appear to be generally sufficient and do not require prohibitive computational overhead.

Given the results in Figure 10, we next wish to evaluate if the proposed sampling scheme enables TANGOS to scale to much bigger models. In order to evaluate this we vary the number of neurons in the relevant hidden layer while maintaining a fixed sampling rate of 50 pairs (consistent with our experiments in Section 5). Other experimental parameters are consistent with the previous experiment. The results are provided in Figure 11 where we observe a relatively slow increase in runtime as the model grows. These results demonstrate that TANGOS can efficiently be applied to much larger models by using our proposed sampling scheme.

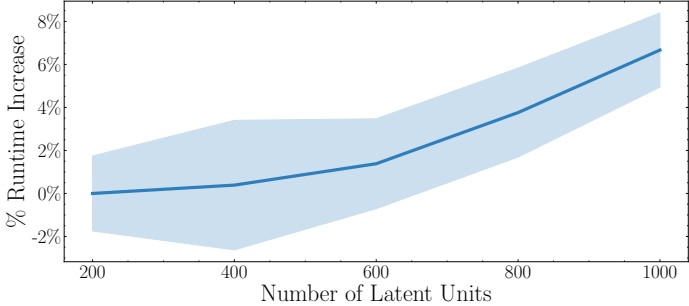

Figure 11: **Scaling to large models.** With a subsampling rate fixed at $M = 50$, TANGOS incurs only a small percentage increase in runtime as the number of neurons in the penultimate hidden layer increases dramatically.

## G ABLATION STUDY

TANGOS is designed with joint application of both regularization on specialization and orthogonalization in mind. Having empirically demonstrated strong overall results, an immediate question is the dynamics of the two regularizers, and how they interact to affect performance. Specifically, we consider the performance gain due to joint regularization effects over applying each regularizer separately.

This includes three separate settings: 1) when the specialization regularizer is applied independently (SpecOnly), here we set $\lambda_2 = 0$ and search over $\lambda_1 \in \{1, 10, 100\}$; 2) when the orthogonalization is applied separately (OrthOnly), we set $\lambda_1 = 0$ and search over $\lambda_2 \in \{0.1, 1\}$; and lastly 3) when both are applied jointly (TANGOS), i.e. searching over $\lambda_1 \in \{1, 10, 100\}$ and $\lambda_2 \in \{0.1, 1\}$. We

report the result of the ablation study in Table 5. We empirically observe that the joint effects of both regularizers (i.e. TANGOS) are crucial to achieve consistently good performance.

Combining these results with what we observed in Figure 3, we hypothesize that applying just specialization regularization, with no regard for diversity, can inadvertently force the neurons to *attend to overlapping regions* in the input space. Correspondingly, simply enforcing orthogonalization, with no regard for sparsity, will likely result in neurons attending to non-overlapping yet *spurious* regions in the input. Thus, we conclude that the two regularizers have distinct, but complementary, effects that work together to achieve the desired regularization effect.

Table 5: **Ablation study.** Generalization performance on different ablation settings.

| Dataset | Classification (Mean NLL) | | | Regression (MSE) | | |
|---|---|---|---|---|---|---|
| | BR | CR | HC | BC | BH | WQ |
| NoReg | 0.0726 | 0.4633 | 0.3321 | 0.3343 | 0.1977 | 0.6732 |
| SpecOnly | 0.0742 | 0.4466 | 0.3837 | 0.3099 | 0.1842 | 0.6714 |
| OrthOnly | 0.0716 | 0.3696 | **0.2073** | 0.2692 | 0.1916 | 0.6529 |
| TANGOS | **0.070** | **0.3633** | 0.2191 | **0.2472** | **0.1826** | **0.6379** |

## H    RANKING PLOT

In Table 1, we reported the generalization performance of TANGOS compared to other regularizers in a stand-alone setting. To gain a better understanding of relative performance, we visually depict the relative ranking of regularizers across all 20 datasets. Figure 12 demonstrates that TANGOS consistently ranks as one of the better-performing regularizers, while performance of benchmark methods tend to fluctuate depending on the dataset.

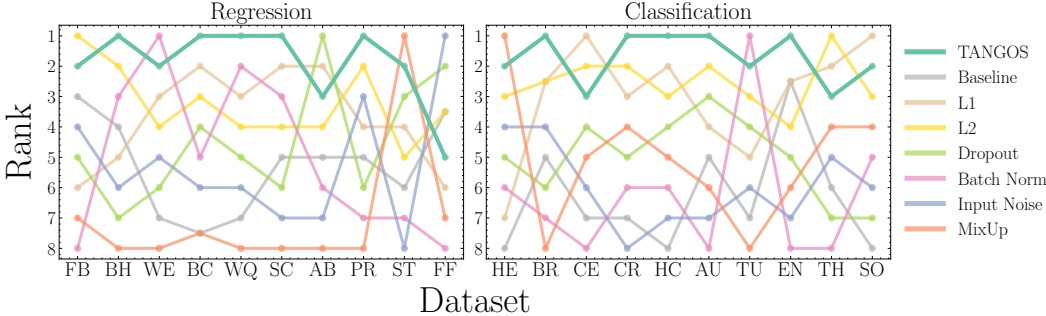

Figure 12: **Ranking of stand-alone regularizers.** Relative ranking of regularizer performance across the 20 datasets, as reported in Table 1. TANGOS consistently ranks among the best-performing regularizers.

## I    STAND-ALONE UNCERTAINTY

In Table 6, we report the standard deviation on generalization performance reported in Table 1. The standard errors are computed using 10 seeded runs.

Table 6: **Standard error on generalization performance.** Standard errors with respect to the random seed after retraining models from Table 1 experiments 10 times.

| Dataset | Baseline | L1 | L2 | DO | BN | IN | MU | TANGOS |
|---------|----------|-----|-----|-----|-----|-----|-----|--------|
| Regression (Mean Squared Error) | | | | | | | | |
| FB | 0.051 | 0.016 | 0.009 | 0.051 | 0.627 | 0.076 | 0.041 | 0.042 |
| BH | 0.023 | 0.021 | 0.029 | 0.022 | 0.025 | 0.023 | 0.011 | 0.028 |
| WE | 0.006 | 0.010 | 0.008 | 0.008 | 0.008 | 0.013 | 0.010 | 0.009 |
| BC | 0.013 | 0.007 | 0.005 | 0.009 | 0.020 | 0.024 | 0.012 | 0.009 |
| WQ | 0.016 | 0.019 | 0.005 | 0.014 | 0.021 | 0.015 | 0.019 | 0.008 |
| SC | 0.026 | 0.014 | 0.019 | 0.025 | 0.023 | 0.168 | 0.067 | 0.017 |
| FF | 0.034 | 0.029 | 0.035 | 0.033 | 0.041 | 0.036 | 0.031 | 0.035 |
| PR | 0.042 | 0.029 | 0.020 | 0.031 | 0.032 | 0.072 | 0.016 | 0.026 |
| ST | 0.090 | 0.084 | 0.085 | 0.064 | 0.076 | 0.080 | 0.082 | 0.029 |
| AB | 0.016 | 0.016 | 0.008 | 0.011 | 0.026 | 0.014 | 0.012 | 0.006 |
| Classification (Mean Negative Log-likelihood) | | | | | | | | |
| HE | 0.057 | 0.049 | 0.009 | 0.033 | 0.163 | 0.038 | 0.067 | 0.032 |
| BR | 0.086 | 0.005 | 0.002 | 0.133 | 0.034 | 0.060 | 0.010 | 0.031 |
| CE | 0.060 | 0.007 | 0.008 | 0.043 | 0.051 | 0.044 | 0.056 | 0.023 |
| CR | 0.029 | 0.094 | 0.004 | 0.034 | 0.041 | 0.027 | 0.027 | 0.019 |
| HC | 0.091 | 0.014 | 0.019 | 0.026 | 0.054 | 0.111 | 0.024 | 0.014 |
| AU | 0.038 | 0.030 | 0.002 | 0.030 | 0.081 | 0.031 | 0.037 | 0.019 |
| TU | 0.075 | 0.075 | 0.087 | 0.077 | 0.087 | 0.096 | 0.078 | 0.064 |
| EN | 0.036 | 0.038 | 0.033 | 0.045 | 0.082 | 0.049 | 0.050 | 0.046 |
| TH | 0.048 | 0.021 | 0.002 | 0.065 | 0.096 | 0.047 | 0.039 | 0.001 |
| SO | 0.038 | 0.028 | 0.016 | 0.046 | 0.029 | 0.025 | 0.023 | 0.008 |

## J    INSIGHTS - EXTENDED RESULTS

In this section, we present extended results of the decomposition of overall model error into diversity and weighted error among an ensemble of latent units from Section 4.2. We include all eight regularizers as described in Table 9 and three datasets (WE, ST, and BC) as described in Table 8. The results are included in Figure 13.

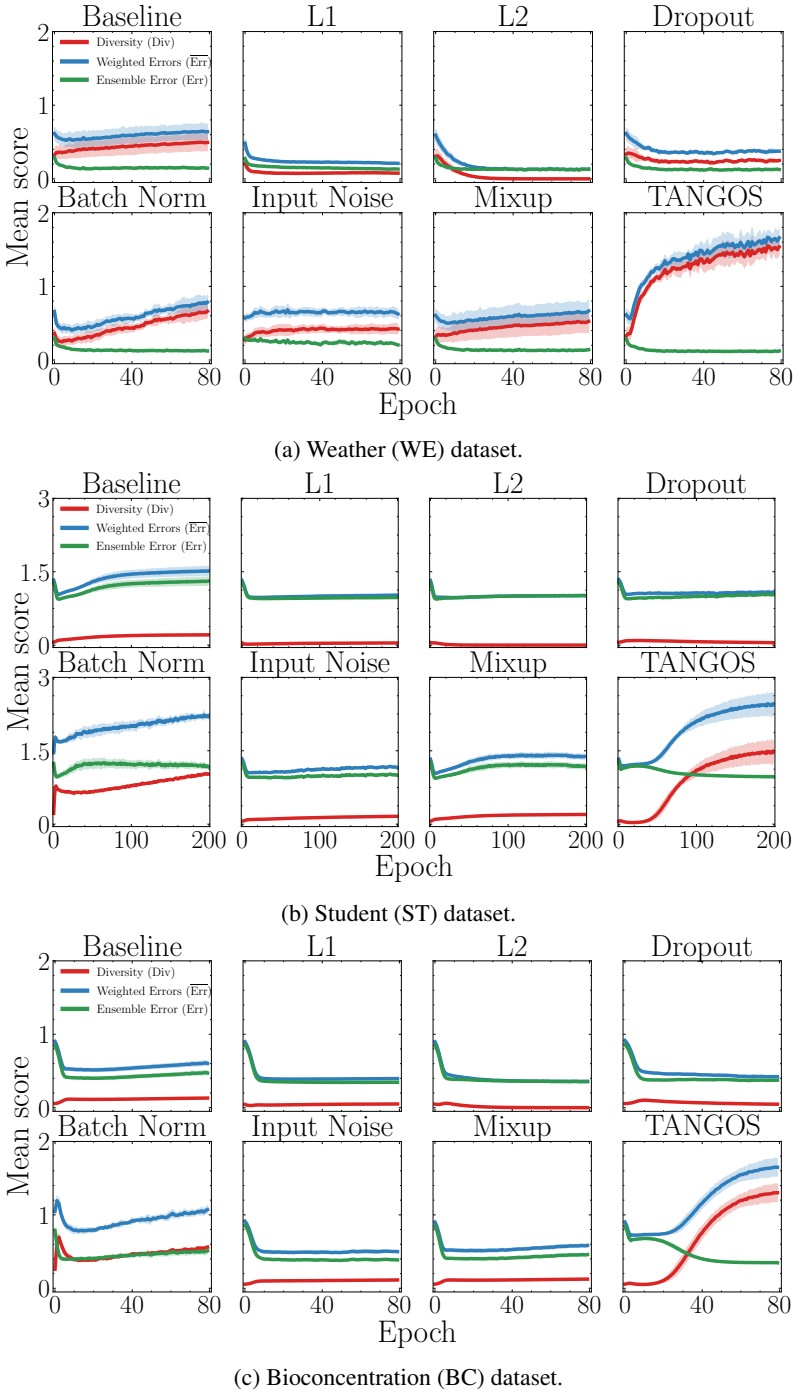

Figure 13: **Neuron diversity.** Further examples of ensemble decomposition $\mathrm{Err} = \overline{\mathrm{Err}} - \mathrm{Div}$ as discussed in Section 4.2.

## K  IN TANDEM RESULTS

In Table 7, we provide a detailed breakdown of Figure 5, specifically by reporting in tandem performance when benchmarks are paired with `TANGOS` across all datasets.

Table 7: **In tandem performance.** Mean $\pm$ standard deviation of generalization performance when each regularizer is employed in tandem with `TANGOS`.

| Dataset | L1 | L2 | DO | BN | IN | MU |
|---|---|---|---|---|---|---|
| Regression (Mean Squared Error) | | | | | | |
| FB | 0.033 $\pm$0.018 | 0.018 $\pm$0.009 | 0.023 $\pm$0.215 | 0.042 $\pm$0.268 | 0.028 $\pm$0.047 | 0.046 $\pm$0.073 |
| BH | 0.192 $\pm$0.022 | 0.176 $\pm$0.024 | 0.196 $\pm$0.024 | 0.220 $\pm$0.021 | 0.191 $\pm$0.023 | 0.178 $\pm$0.019 |
| WE | 0.093 $\pm$0.011 | 0.091 $\pm$0.009 | 0.092 $\pm$0.009 | 0.076 $\pm$0.009 | 0.081 $\pm$0.012 | 0.077 $\pm$0.013 |
| WQ | 0.637 $\pm$0.014 | 0.639 $\pm$0.006 | 0.628 $\pm$0.008 | 0.630 $\pm$0.015 | 0.644 $\pm$0.011 | 0.649 $\pm$0.018 |
| BC | 0.227 $\pm$0.011 | 0.236 $\pm$0.011 | 0.243 $\pm$0.013 | 0.275 $\pm$0.027 | 0.235 $\pm$0.009 | 0.260 $\pm$0.014 |
| SC | 0.407 $\pm$0.044 | 0.399 $\pm$0.072 | 0.370 $\pm$0.031 | 0.408 $\pm$0.049 | 0.391 $\pm$0.317 | 0.425 $\pm$0.061 |
| AB | 0.309 $\pm$0.028 | 0.312 $\pm$0.028 | 0.312 $\pm$0.026 | 0.311 $\pm$0.027 | 0.319 $\pm$0.037 | 0.308 $\pm$0.03 |
| FF | 1.281 $\pm$0.029 | 1.276 $\pm$0.043 | 1.268 $\pm$0.028 | 1.297 $\pm$0.027 | 1.203 $\pm$0.046 | 1.207 $\pm$0.011 |
| PR | 0.553 $\pm$0.047 | 0.572 $\pm$0.031 | 0.642 $\pm$0.093 | 0.561 $\pm$0.069 | 0.565 $\pm$0.025 | 0.568 $\pm$0.081 |
| ST | 0.392 $\pm$0.006 | 0.382 $\pm$0.006 | 0.388 $\pm$0.016 | 0.446 $\pm$0.018 | 0.417 $\pm$0.012 | 0.447 $\pm$0.011 |
| Classification (Mean Negative Log-likelihood) | | | | | | |
| HE | 0.441 $\pm$0.046 | 0.427 $\pm$0.049 | 0.454 $\pm$0.095 | 0.407 $\pm$0.047 | 0.377 $\pm$0.067 | 0.397 $\pm$0.075 |
| BR | 0.074 $\pm$0.005 | 0.070 $\pm$0.002 | 0.068 $\pm$0.006 | 0.062 $\pm$0.003 | 0.065 $\pm$0.01 | 0.078 $\pm$0.011 |
| CE | 0.389 $\pm$0.007 | 0.396 $\pm$0.007 | 0.394 $\pm$0.033 | 0.446 $\pm$0.024 | 0.422 $\pm$0.075 | 0.422 $\pm$0.062 |
| CR | 0.362 $\pm$0.021 | 0.366 $\pm$0.015 | 0.364 $\pm$0.003 | 0.406 $\pm$0.023 | 0.367 $\pm$0.01 | 0.384 $\pm$0.036 |
| HC | 0.200 $\pm$0.056 | 0.179 $\pm$0.012 | 0.185 $\pm$0.009 | 0.186 $\pm$0.021 | 0.211 $\pm$0.008 | 0.181 $\pm$0.037 |
| AU | 0.368 $\pm$0.028 | 0.360 $\pm$0.138 | 0.352 $\pm$0.011 | 0.344 $\pm$0.013 | 0.379 $\pm$0.016 | 0.368 $\pm$0.041 |
| TU | 1.519 $\pm$0.082 | 1.506 $\pm$0.045 | 1.481 $\pm$0.048 | 1.506 $\pm$0.067 | 1.522 $\pm$0.049 | 1.503 $\pm$0.087 |
| SO | 0.227 $\pm$0.036 | 0.268 $\pm$0.071 | 0.233 $\pm$0.021 | 0.353 $\pm$0.052 | 0.257 $\pm$0.032 | 0.304 $\pm$0.054 |
| EN | 0.990 $\pm$0.024 | 0.971 $\pm$0.002 | 0.945 $\pm$0.001 | 1.004 $\pm$0.026 | 0.995 $\pm$0.001 | 1.007 $\pm$0.04 |
| TH | 0.506 $\pm$0.031 | 0.503 $\pm$0.008 | 0.513 $\pm$0.01 | 0.524 $\pm$0.008 | 0.512 $\pm$0.045 | 0.514 $\pm$0.03 |

## L  DATASET AND REGULARIZER DETAILS

We perform our experiments on 20 real-world publicly available datasets obtained from (Dua et al., 2017). They are summarized in Table 8. Further information and the source files used for each of the respective datasets can be found at: `https://archive.ics.uci.edu/ml/machine-learning-databases/<UCISource>/` where `<UCI Source>` denotes the datasets unique identifier as listed in Table 8. Standard preprocessing was applied including standardization of features, one hot encoding of categorical variables, median imputation of missing values and log transformations of highly skewed feature distributions. Furthermore, for computational feasibility, datasets with over 1000 samples were reduced in size. In these cases the first 1000 samples from the original UCI Source file were used. In Table 9 we summarize the regularizers considered in this work.

Table 8: **Dataset descriptions.** Summary of the datasets considered in this work.

| Dataset | UCI Source | Type | Feature size | Sample Size | Reference |
|---|---|---|---|---|---|
| Facebook (FB) | "00368" | Regression | 21 | 495 | Moro et al. (2016) |
| Boston (BH) | [1] | Regression | 13 | 506 | Harrison Jr & Rubinfeld (1978) |
| Weather (WE) | "00514" | Regression | 45 | 1000 | Cho et al. (2020) |
| Wine Quality (WQ) | "wine-quality" | Regression | 11 | 1000 | Cortez et al. (2009) |
| Bioconcentration (BC) | "00510" | Regression | 45 | 779 | Grisoni et al. (2015) |
| Skillcraft (SC) | "00272" | Regression | 18 | 1000 | Thompson et al. (2013) |
| Forest Fire (FF) | "forest-fires" | Regression | 39 | 517 | Cortez & Morais (2007) |
| Protein (PR) | "00265" | Regression | 9 | 1000 | Dua et al. (2017) |
| Student (ST) | "00320" | Regression | 56 | 649 | Cortez & Silva (2008) |
| Abalone (AB) | "abalone" | Regression | 9 | 1000 | Waugh (1995) |
| Heart (HE) | "statlog" | Classification | 20 | 270 | Dua et al. (2017) |
| Breast (BR) | "breast-cancer-wisconsin" | Classification | 9 | 699 | Street et al. (1993) |
| Cervical (CE) | "00383" | Classification | 136 | 858 | Fernandes et al. (2017) |
| Credit (CR) | "credit-screening" | Classification | 40 | 677 | Dua et al. (2017) |
| HCV (HC) | "00571" | Classification | 12 | 615 | Hoffmann et al. (2018) |
| Australian (AU) | "statlog" | Classification | 55 | 690 | Quinlan (1987) |
| Tumor (TU) | "primary-tumor" | Classification | 25 | 339 | Michalski et al. (1986) |
| Entrance (EN) | "00582" | Classification | 38 | 666 | Hussain et al. (2018) |
| Thoracic (TH) | "00277" | Classification | 24 | 470 | Zikeba et al. (2013) |
| Soybean (SO) | "soybean" | Classification | 484 | 683 | Fisher & Schlimmer (1988) |

[1] This dataset has now been removed due to ethical issues. For more information see the following url
`https://medium.com/@docintangible/racist-data-destruction-113e3eff54a8`

Table 9: **Overview of regularizers.** Description of benchmarks considered in this work and their implementations.

| Regularizer | Reference | Implementation |
|---|---|---|
| Baseline | NA | Paszke et al. (2019) |
| L1 | Tibshirani (1996) | Paszke et al. (2019) |
| L2 | Hoerl & Kennard (1970) | Paszke et al. (2019) |
| Dropout | Hinton et al. (2012) | Paszke et al. (2019) |
| Batch Norm | Ioffe & Szegedy (2015) | Paszke et al. (2019) |
| Input Noise | Krizhevsky et al. (2012) | Paszke et al. (2019) |
| Mixup | Zhang et al. (2018) | Zhang et al. (2018) |
| TANGOS | This work | Paszke et al. (2019) |

