# OpenReview forum: "TANGOS: Regularizing Tabular Neural Networks through Gradient Orthogonalization and Specialization"
_ICLR.cc/2023/Conference — ICLR 2023 poster_

### Official Review · Reviewer_qYko · 2022-10-24

**Confidence:** 4
**Correctness:** 3
**Technical Novelty And Significance:** 2
**Empirical Novelty And Significance:** 2
**Recommendation:** 5

**Clarity, Quality, Novelty And Reproducibility:**

Paper is well written and the core idea is clear.

Quality is mediocre. Experiments section can be improved.

I think Novelty is low. Methodological innovations are relatively straightforward

Reproducibility: Some details are left out to fully reproduce.

**Strength And Weaknesses:**

Strengths

- Extensive analyses on the working principles of the method

- Empirical results are showing consistent improvements

- Insights on why the method works

Weaknesses

- The contributions are small - specialization and orhthonogalization are straightforward ideas.

- Overall, very small and simple datasets are used in experiments. Real-world tabular data would have more samples with more complex dynamics.

- The improvements are very small - the results are barely better than L2 regularization.

- The impact is not analyzed with multiple commonly used tabular deep learning architectures.

- It is unclear whether the hyperparameters are reoptimized for each method.

- No experimental results on randomness, showing error bars at the main events could be useful.

**Summary Of The Paper:**

The paper introduces TANGOS, a novel regularization method which promotes specialization and orthogonalization among the gradient attributions of the latent units of a neural network. TANGOS has benefits for out-of-distribution generalization and can be combined with others. Extensive empirical evaluation with TANGOS is presented.

**Summary Of The Review:**

The paper is borderline. In general, the empirical validation is weak and the contributions are not major.

---

> ### Author Response · Authors · 2022-11-14
> **Response 1 to reviewer qYko**
>
> We thank reviewer qYko for taking the time to review our work.
>
> **P1** _The contributions are small - specialization and orhthonogalization are straightforward ideas._
>
> We begin by addressing the claim that the ideas used in this paper are “straightforward”. In fact, we agree that the ideas presented in this paper are conceptually intuitive. However, we do not view this as a weakness. Simple and effective ideas are often more easily understood and adapted in practice. Throughout the sciences, simple theories are generally sought after and in machine learning research we would argue that many of the most impactful developments have been relatively straightforward in retrospect (e.g. dropout, batch-norm and input noise). Indeed, even backpropagation, the fundamental learning mechanism of neural networks, is a straightforward application of the chain rule of calculus. We also hope that the key ideas of this paper appearing straightforward may be indicative of a well-written paper (we note that all three reviewers commented positively on the presentation of this work in terms of writing/presentation/clarity).
>
> However, we disagree that this would imply the contributions of this work are necessarily small. In this work, we have proposed a general-purpose regularization method, which we demonstrate to be effective in the tabular domain. In this domain, **we regularly outperform some of the most commonly used regularizers including L2/weight decay, dropout, and batch normalization**. Since these are the standard methods used by machine learning practitioners, proposing a method which can often outperform them is a highly significant contribution. Furthermore, we would **highlight that tabular data is significantly understudied relative to its ubiquity in real-world data**. In fact, the top 50 keywords from the ICLR 2022 conference didn’t reference tabular data (or equivalent) while images, time series and natural language processing all featured [1]. In contrast, tabular data constitutes the majority of real-world data science applications. A recent Kaggle survey found that among a sample of 14,000 practising data scientists, 65% reported working with tabular-style data on a daily basis [2,3]. This can be compared to the 14% that reported working with image data daily. We would therefore claim that the growing number of contributions in this area are of particular significance.
>
> [1] https://github.com/fedebotu/ICLR2022-OpenReviewData
>
> [2] Sun, B., Yang, L., Zhang, W., Lin, M., Dong, P., Young, C. and Dong, J., 2019. Supertml: Two-dimensional word embedding for the precognition on structured tabular data. In Proceedings of the IEEE/CVF Conference on Computer Vision and Pattern Recognition Workshops (pp. 0-0).
>
> [3] Kaggle Machine Learning & Data Science Survey, 2017

---

> ### Author Response · Authors · 2022-11-14
> **Response 2 to reviewer qYko**
>
> **P2** _Overall, very small and simple datasets are used in experiments. Real-world tabular data would have more samples with more complex dynamics._
>
> While we understand the reviewer's remark, we would like to argue that the datasets used in our work are not simple and are quite representative of challenges that machine learning practitioners face in applications such as healthcare, finance, insurance, climate science, and sport science. Furthermore, **when learning on tabular data**, where we typically don’t have strong priors on the data distribution (as opposed to e.g. images) and where datasets are typically smaller, **appropriate regularization plays a particularly significant role** in shaping inductive biases and generalizing to unseen data. However, we agree that a broader evaluation of a method is always useful and therefore have added a **new experiment with two additional large datasets in Appendix I** consisting of 83733 and 98050 examples. Consistent with our existing findings, we again find that TANGOS is effective in this experiment.
>
> In order to demonstrate the complexity and real-world nature of the datasets used in this work, in what follows we provide some further discussion on them and their background. The 20 datasets used in this work have all been used in previous academic research and were selected to characterize the broad selection of data faced by practising statisticians, data scientists and researchers. In the following four paragraphs, we provide some additional background information on a sample of these datasets to highlight both their practical significance and the considerable complexity in achieving a highly performant model.
>
> * **The tumor dataset** (TU) [1] requires the learning algorithm to distinguish between 22 possible locations of a primary tumor based on a 1986 dataset of patients with known tumor locations. A clinician's accurate diagnosis of the location of a tumor can result in faster treatment for the patient and can prevent incorrect resource allocation for the hospital. However, this task is challenging with specialist oncologists only achieving 42% accuracy on the test set while internists only achieve 32%.
>
> * **The forest fire dataset** (FF) [2] was collected from the Montesinho natural park in Portugal between January 2000 and December 2003. The information collected consists of a broad range of factors considered by domain experts to be useful in the prediction of future forest fires. As noted in the paper, “forest fires are a major environmental issue, creating economical and ecological damage while endangering human lives. Fast detection is a key element for controlling such phenomenon”. In climate science, this problem is more relevant than ever with fast response essential to saving lives in affected areas. However, when evaluated, the most accurate models only achieve 46% accuracy in predicting fire damage on fine-grained areas of 1ha.
>
> * **The Facebook metrics dataset** (FB) [3] contains information on all of the posts published by a world-renowned cosmetics brand in the year 2014 on Facebook. As noted in the paper “Defining a causal relation between the knowledge found and brand building by relating the influence of the input features and the impact of the posts on customers, and hypothesizing on how such metrics can effectively contribute to brand recognition”. However, modelling this data is challenging due largely to a low signal-to-noise ratio. Even hand-crafted state-of-the-art approaches were only able to achieve 27% mean absolute error on the test set.
>
> * **The thoracic dataset** (TH) [4] was collected from patients in Wroclaw Thoracic Surgery Centre, Poland in order to develop improved methods for patient selection for thoracic surgery. This decision must consider the likelihood of post-operative complications as well as a longer-term prosepective based on a large selection of patient characteristics. In practice, this high-stakes decision must be made under significant uncertainty with even the best-performing models only able to achieve a 60% true positive rate and 72% false negative rate when evaluated on test data.
>
>
> [1] Ryszard S Michalski, Igor Mozetic, Jiarong Hong, and Nada Lavrac. The multi-purpose incremental learning system aq15 and its testing application to three medical domains. In Proc. AAAI, volume 1986, pp. 1–041, 1986.
>
> [2] Paulo Cortez and Aníbal de Jesus Raimundo Morais. A data mining approach to predict forest fires using meteorological data. 2007.
>
> [3] Sérgio Moro, Paulo Rita, and Bernardo Vala. Predicting social media performance metrics and evaluation of the impact on brand building: A data mining approach. Journal of Business Research, 69(9):3341–3351, 2016.
>
> [4] Maciej Zikeba, Jakub M Tomczak, Marek Lubicz, and Jerzy ’Swikatek. Boosted svm for extracting rules from imbalanced data in application to prediction of the post-operative life expectancy in the lung cancer patients. Applied Soft Computing, 2013.

---

> > ### Comment · Reviewer_qYko · 2022-11-15
> > **Datasets**
> >
> > All of these datasets have <1000 samples. They are from aged papers. With modern data infrastructure, the datasets in such applications can be much much larger. There are datasets from Healthcare or Tech industry that are orders of magnitude larger than these.
> >
> > On two new larger datasets, which are still not very large with <100k samples, the improvement of TANGOS is quite small, and deep learning with regularization is underperformed by ensemble decision trees.

---

> > > ### Author Response · Authors · 2022-11-18
> > > **Response to both comments linked below**
> > >
> > > We thank the reviewer for this comment. We have responded to these two new comments together at the following link.
> > >
> > > https://openreview.net/forum?id=n6H86gW8u0d&noteId=KvQ0kaqGCW

---

> ### Author Response · Authors · 2022-11-14
> **Response 3 to reviewer qYko**
>
> **P3** _The improvements are very small - the results are barely better than L2 regularization._
>
> Thank you for this comment. We want to highlight that 1. L2 regularization is practically the most used regularization method in deep learning, and 2. **out-performing L2 regularization in the tabular domain, even by a small margin, is a valuable contribution**. Unlike image and sequence data, tabular data is often characterized by noisy observations and a relatively small sample size. Often, this means that DL models rely heavily on regularization to induce the correct inductive biases to generalize well to unseen data [1].
>
> With this in mind, we also wish to point out the set of in-tandem results (see Section 5.2). Our goal here was to highlight that, by regularizing a distinct objective, our method nicely complements popular forms of regularization. Therefore, by employing regularizers in tandem, we can further improve performance.
>
> [1] Kadra, A., Lindauer, M., Hutter, F. and Grabocka, J., 2020. Regularization Cocktails for Tabular Datasets.
>
>
> **P4** _The impact is not analyzed with multiple commonly used tabular deep learning architectures._
>
> We thank the reviewer for raising this. Although we believe the experiments we ran across 20 tabular datasets to be a thorough investigation of the method, we agree that some investigation of alternative architectures will significantly improve the paper. **We have therefore added two new experiments that apply TANGOS on two very different architectures**. We summarize these experiments below and have updated the manuscript to include these experiments.
>
> (1) FT-Transformer - A feature tokenizer transformer architecture was recently proposed as a state-of-the-art deep learning architecture for tabular data in [1]. Specifically, this architecture combines a Feature Tokenizer which transforms features into embeddings with a multi-layer Transformer [2]. In our experiments, we use a 3-layer Transformer with a 32-dimensional feature embedding size and 4 attention heads. Following the original paper, we use Reglu activations, a hidden layer size of 43 corresponding to a ratio of 4/3 with the embedding size, Kaiming initialization [3], and AdamW optimizer [4]. Finally, we apply a learning rate of 0.001. We compare this architecture with and without TANGOS regularization applied. **We found TANGOS to be effective in this experiment**. See a tabular summary of the baseline vs TANGOS results in [this table](https://imgur.com/a/0p4ibcH) and the **full experiment description and results in Appendix I**.
>
> (2) Convolutional Neural Network - **In Appendix H we have added a behavioral analysis of TANGOS using a convolutional neural network**. This experiment was designed to provide further insights into how a TANGOS regularized network differs from existing training methods by examining the input gradients corresponding to each of the hidden units in the penultimate layer. We found that, **in contrast to the baseline training methods, the TANGOS regularized models latent units attend to sparse and orthogonal regions in the input space**. As an illustration, we interpret what these latent units are attending to across a range of samples and often observe intuitive behavior in which latent units attend to sensible features. For example, in the updated manuscript, we highlight a neuron that appears to discriminate between an open or a closed loop at the lower left of the digit. This is a key aspect of distinction between the set of digits {2, 6, 8, 0} and {9, 5, 3}. We consider further development of these methods for additional domains and/or more interpretable models to be an interesting direction for future work.
>
> [1] Gorishniy, Yury, et al. "Revisiting deep learning models for tabular data." Advances in Neural Information Processing Systems 34 (2021)
>
> [2] Ashish Vaswani, Noam Shazeer, Niki Parmar, Jakob Uszkoreit, Llion Jones, Aidan N Gomez, Łukasz Kaiser, and Illia Polosukhin. Attention is all you need. Advances in neural information processing systems, 30, 2017.
>
> [3] Kaiming He, Xiangyu Zhang, Shaoqing Ren, and Jian Sun. Delving deep into rectifiers: Surpassing human-level performance on imagenet classification. In Proceedings of the IEEE international conference on computer vision, pp. 1026–1034, 2015.
>
> [4] Ilya Loshchilov and Frank Hutter. Decoupled weight decay regularization. arXiv preprint
> arXiv:1711.05101, 2017.

---

> > ### Comment · Reviewer_qYko · 2022-11-15
> > **Response to rebuttal**
> >
> > "Unlike image and sequence data, tabular data is often characterized by noisy observations and a relatively small sample size. " - This claim is not always true. There can be tabular datasets with billions of samples or without any noisy observations. Also the paper does not analyze where the improvements come from, e.g. whether they are bigger for noisy features or labels.
> >
> > There are many other papers like "[1] Kadra, A., Lindauer, M., Hutter, F. and Grabocka, J., 2020. Regularization Cocktails for Tabular Datasets." that show regularization is quite useful on some datasets, so it is not a new observation.

---

> > > ### Author Response · Authors · 2022-11-18
> > > **Response to comment [1/2]**
> > >
> > > Thank you for your response.
> > >
> > > We did not claim it to be **always** true, we stated it was **often** true. We don’t doubt the existence of such large tabular datasets, but they are far from standard. We refer again to the same Kaggle survey [1] of over 1600 practicing data scientists where only 2.7% of those working with tabular data reported working with large-scale data ($>1$TB) while **65.7% of that same group reported that they typically work with smaller data** ($\leq1$GB).
> > >
> > > We thank the reviewer for raising the example of the Kadra et. al. paper. This is the arxiv version of the same paper [2] which we discussed in our manuscript and also referenced in our previous rebuttal to this reviewer (see P3). In fact, the results of this paper illustrate the point we are making, especially with respect to the points raised by this reviewer in their other comment which we quote below.
> > >
> > > _"On two new larger datasets, which are still not very large with <100k samples, the improvement of TANGOS is quite small, and deep learning with regularization is underperformed by ensemble decision trees."_
> > >
> > > Actually, we believe the Kadra et. al. paper [2] illustrates our points well. Firstly, we note that **of the 40 datasets used in their experiments, only 8 contained >100,000 examples** with the largest containing 416,188 examples. Secondly, **this paper shows that we should only expect relatively small gains from regularization on large tabular datasets**. The work of Kadra et. al. proposed to simultaneously optimize 14 different methods of regularization using Bayesian optimization on a single neural network, seeking to extract every possible gain from regularization. However, when we examine the reported results (Table 2) obtained on these datasets, namely: clickpred, ccfraud, aloi, dionis, miniboone, Idpa and walking (we exclude skin-seg which failed to learn), we find that the improvements are marginal. **Three datasets in fact perform worse, and of the four that do perform better, it is by an average improvement of 1% accuracy**. This does not invalidate the methods of Kadra et. al., rather it highlights that we should be realistic in our expectations of magnitudes in performance gains.
> > >
> > > However, we do agree that it is valuable to analyze how a method's performance is affected by dataset size. Therefore, **we have added a new experiment (Appendix J) where we analyze TANGOS performance on Dionis, the largest of the 40 benchmark datasets proposed in the Kadra et. al. paper with 416,188 examples**. We consider training a model on fractions of the training data (10%, 50%, 100%) with and without TANGOS regularization. We also compare to the best performing baseline, L2 regularization. At each fraction, we train three TANGOS combinations and three L2 combinations and select the best-performing model on validation data. We repeat this procedure for six runs. **We find that TANGOS achieves significantly better performance at all fractions of the data** when compared to both the unregularized model and L2. In other words, the performance gain from TANGOS regularization is reflected at smaller and larger sample sizes. We have included a full description of this experiment in the manuscript, as well as the main results [here](https://imgur.com/a/Bf86nlN).
> > >
> > > Additionally, to further investigate the magnitude of TANGOS improvement in our Section 5.1 experiments, **we perform a Wilcoxon signed-rank sum test [3] comparing TANGOS to the best-performing baseline method** (L2). We perform the test on both the regression and classification results and obtain p-values of 0.006 and 0.026 respectively, which can be interpreted as **strong evidence to suggest TANGOS gains are statistically significant** in both cases. **We have added these results to the manuscript**.
> > >
> > > _continued in next comment_

---

> > > > ### Author Response · Authors · 2022-11-18
> > > > **Response to comment [2/2]**
> > > >
> > > > _continued from the previous comment_
> > > >
> > > > In your other comment you said:
> > > >
> > > > _They are from aged papers. With modern data infrastructure, the datasets in such applications can be much much larger. There are datasets from Healthcare or Tech industry that are orders of magnitude larger than these._
> > > >
> > > > Some of the datasets used are older while 10 are published in the last 7 years. However, we fail to see an issue with including older datasets too. As we already detailed, datasets such as the tumor (estimating tumor location from patient characteristics) and forest fire (predicting risk of forest fire) are just as relevant and challenging now as when they were first published. This reviewer questioned the real-world nature and complexity of these datasets, and **we believe they make excellent benchmark tasks representing exactly the type of applications faced by practitioners for which we should strive to develop machine learning methods**.
> > > >
> > > > Regarding the quality, complexity and value of the datasets used in our experiments we highlight that:
> > > >
> > > > (a) **The datasets used in this work are regularly used as benchmarks throughout the machine learning literature**. For example: from just papers we are already familiar with published in the past 12 months alone one can find BR, CR, HE, PR, AU in [4], AB, WQ, BR in [5], SC in [6] and AB, CR, BR in [7]. However, this is just a tiny subset with these datasets cited in 100’s of papers every year.
> > > >
> > > > (b) **These datasets are also important within their respective fields**. For example: the FB dataset is regularly analysed in the business research and marketing community (e.g. [10,11]), a recent journal article in the healthcare domain explores neural network solutions specifically on the TH dataset [9], and a 2021 survey of machine learning algorithms based forest fires prediction and detection systems [8] dedicated an entire subsection to FF.
> > > >
> > > >
> > > >
> > > > [1] Kaggle Machine Learning & Data Science Survey, 2017.
> > > >
> > > > [2] Kadra, A., Lindauer, M., Hutter, F. and Grabocka, J., 2020. Regularization Cocktails for Tabular Datasets.
> > > >
> > > > [3] Wilcoxon, Frank. "Individual comparisons by ranking methods." Breakthroughs in statistics. Springer, New York, NY, 1992. 196-202.
> > > >
> > > > [4] Hollmann, Noah, et al. "TabPFN: A Transformer That Solves Small Tabular Classification Problems in a Second." arXiv preprint arXiv:2207.01848 (2022).
> > > >
> > > > [5] Shenkar, Tom, and Lior Wolf. "Anomaly detection for tabular data with internal contrastive learning." International Conference on Learning Representations. 2021.
> > > >
> > > > [6] Yu, Haolin, et al. "Federated Bayesian Neural Regression: A Scalable Global Federated Gaussian Process." arXiv preprint arXiv:2206.06357 (2022).
> > > >
> > > > [7] Borgohain, Satya, Klaus Ackermann, and Ruben Loaiza-Maya. "Bayesian Neural Network Versus Ex-Post Calibration For Prediction Uncertainty." arXiv preprint arXiv:2209.14594 (2022).
> > > >
> > > > [8] Abid, Faroudja. "A survey of machine learning algorithms based forest fires prediction and detection systems." Fire Technology 57.2 (2021): 559-590.
> > > >
> > > > [9] Ravichandran, Akshaya, et al. "Post Thoracic Surgery Life Expectancy Prediction Using Machine Learning." International Journal of Healthcare Information Systems and Informatics (IJHISI) 16.4 (2021): 1-20.
> > > >
> > > > [10] Shahbaznezhad, Hamidreza, Rebecca Dolan, and Mona Rashidirad. "The role of social media content format and platform in users' engagement behavior." Journal of Interactive Marketing 53 (2021): 47-65.
> > > >
> > > > [11] Schaefers, Tobias, et al. "More of the same? Effects of volume and variety of social media brand engagement behavior." Journal of business research 135 (2021): 282-294.

---

> > > ### Author Response · Authors · 2022-12-05
> > > **Follow up**
> > >
> > > We thank the reviewer again for their feedback and wish to follow up to ensure that we have addressed all points to their satisfaction.
> > > As can be seen from the updates we have made to the manuscript, we have carefully considered your feedback and made significant improvements to the paper. These changes are summarized in our responses [on OpenReview](https://openreview.net/forum?id=n6H86gW8u0d&noteId=lgxdbkhHPW). We also wish to highlight our discussions with the other reviewers who were both positive about the paper. We hope these dialogues might also help allay any outstanding concerns the reviewer may have that are currently impeding recommending acceptance.
> > >
> > > We hope that our responses and the updates we have made to the manuscript have addressed your concerns and convinced you of the value and importance of our work. If you have any further questions or comments, please do not hesitate to contact us. We would be happy to discuss the paper with you further prior to the end of the discussion period on December 12th.

---

> ### Author Response · Authors · 2022-11-14
> **Response 4 to reviewer qYko**
>
> **P5** _It is unclear whether the hyperparameters are reoptimized for each method._
>
> We would like to clarify that the **hyperparameters were indeed reoptimized for each method**. As stated in the paper “in all experiments, we use 5-fold cross validation to train and validate each benchmark. We select the model which achieves the lowest validation error and provide a final evaluation on a held-out test set.” (see the last two sentences in Section 5). This is in line with best practices, where we use validation data to select the best-performing hyperparameters (for each dataset, and each method, separately), before reporting the performance of a single evaluation on the test set. This protocol was repeated in its entirety for each run.
>
>
>
> **P6** _No experimental results on randomness, showing error bars at the main events, could be useful._
>
> We would like to point out that **standard errors on all results are indeed available in the paper**, which is mentioned in the main paper: “There we also provide a similar table displaying standard errors.” For the results in the figures, we visualized the standard deviation over ten seeded runs as shaded regions on the plot. Similarly, we reported numeric standard deviation on results in Tables 2 and 3 in Appendices B and D respectively.
>
> **P7** _Reproducibility: Some details are left out to fully reproduce_
>
> This comment is surprising to us as **we believe we included all experimental details in the paper** in addition to **the code used to produce the main results in the supplementary material** as well as a brief description of how to run our experiments. Furthermore, both reviewers xMqZ and qYko explicitly commented on the fact that we did provide sufficient detail to reproduce our results. However, if this reviewer has noticed any specific missing details we would be happy to update the manuscript accordingly.

---

> ### Author Response · Authors · 2022-11-14
> **Response 5 to reviewer qYko**
>
> **P8** _I think Novelty is low._
>
> We thank the reviewer for raising this question, however, we would like to argue that our treatment of specialization and orthogonalization is novel and brings a significant contribution to the literature. We split our discussion into two key aspects of novelty: literature novelty and behavioral novelty.
>
> **Literature novelty** - Here we refer to the novelty of our method with respect to existing works in the literature. A number of previous works have considered using gradient attributions to improve model robustness to input perturbations (e.g. [2]) which is a useful defense against some adversarial attacks [3]. Typically, these methods use gradients with respect to the output layer rather than a hidden layer, in contrast to this work, as they wish to permute inputs while minimizing or maximizing the change in target. Furthermore, **invariance to small perturbations of inputs is often less desirable in the tabular setting** than in e.g. images as one pixel is unlikely to change the class of an image while one important feature can significantly change the classification of a patient record or a credit decision. Another line of research uses input gradients to manually enforce a network to attend to human-crafted regions in the input (e.g. [4,5]). **This requires a manually annotated region for the network to attend** which is clearly unrelated to the method presented in TANGOS. Finally, [6] considers using gradient attributions as priors to encourage certain properties such as smoothness between pixels in an image. Again, this work is in the context of gradients **with respect to the output layer** and therefore **does not consider the relationship between gradient attributions of hidden units**. We hope that this summary further demonstrates the novelty of this work with respect to existing literature. If the reviewer is aware of any specific existing works that deal with specialization and orthogonalization between the gradient attributions of latent units in neural networks we would be happy to reevaluate our position on this issue.
>
> **Behavioral novelty** - Here we refer to the novel behaviour induced in the network when compared to existing models. TANGOS regularization results in a model that **behaves very differently to existing methods of training**. In section 4.1 we showed that TANGOS does indeed result in models that attend to specialised and orthogonal regions in the input space as desired. This was in contrast to existing methods which achieve improved performance by other means. In section 4.2 we applied simple ensemble theory to show that TANGOS regularization results in a model in which the latent units behave more like a functioning ensemble. Again this was in contrast to existing regularization methods. In addition to these points, **we have added a new experiment in Appendix H in which we perform an illustrative analysis of exactly how a TANGOS regularized model is distinct from standard training**. In order to maximise the interpretability of this experiment, we evaluate TANGOS on a simple image classification task with the MNIST dataset. By comparing the gradient attributions of the TANGOS-trained model with those of a baseline, we highlight that our method results in specialised and decorrelated latent units. We also find that in many cases we can provide a clear interpretation of the roles played by each latent unit which provides an exciting direction for future research.
>
> To summarize, we believe this work to be novel with respect to existing literature as well as the behavior it induces in the network. We thank this reviewer and **we have added some minor clarification to reflect this discussion in the related works section of the main text**.
>
> [1] Higgins, Irina, et al. "beta-vae: Learning basic visual concepts with a constrained variational framework." (2016).
>
> [2] Drucker, H. and Le Cun, Y. (1992). Improving generalization performance using double backpropa- gation. IEEE Transactions on Neural Networks, 3(6):991–997.
>
> [3] Moosavi-Dezfooli, S.-M., Fawzi, A., Uesato, J., and Frossard, P. (2019). Robustness via curvature regularization, and vice versa. In Proceedings of the IEEE/CVF Conference on Computer Vision and Pattern Recognition, pages 9078–9086.
>
> [4] Liu, F. and Avci, B. (2019). Incorporating priors with feature attribution on text classification.
>
> [5] Chen, J., Wu, X., Rastogi, V., Liang, Y., and Jha, S. (2019). Robust attribution regularization. Advances in Neural Information Processing Systems, 32.
>
> [6] Erion, G., Janizek, J. D., Sturmfels, P., Lundberg, S. M., and Lee, S.-I. (2021). Improving performance of deep learning models with axiomatic attribution priors and expected gradients. Nature machine intelligence, 3(7):620–631.

---

### Official Review · Reviewer_Aiai · 2022-10-26

**Confidence:** 3
**Correctness:** 3
**Technical Novelty And Significance:** 3
**Empirical Novelty And Significance:** 3
**Recommendation:** 8

**Clarity, Quality, Novelty And Reproducibility:**

The paper's writing and presentation of the work is clear and the results should be reproducible. As mentioned early, the novelty is the limited aspect.

**Strength And Weaknesses:**

Strength:
1. The experiments has been conducted very extensively to demonstrate the effectiveness of the method.
2. The idea makes intuitively senses.

Weakness:
1. The idea novelty is limited. The specialization and orthogonalization idea are not that new.
2. There is no ablation study on which part is more important, specialization or orthogonalization.

**Summary Of The Paper:**

The paper propose a new regularization technique for the structured tabular data, namely TANGOS. The method works because intuitively it makes sense to enforce specialization of each unit and orthogonalization between units. The paper has also done extensive experiments on regression and classification to support the effectiveness of the regularization method.

**Summary Of The Review:**

Overall, the paper proposes an interesting and effective regularization method. The experiments have been done extensively. Despite the limitation of the novelty, the paper is still a well-written paper.

---

> ### Author Response · Authors · 2022-11-14
> **Response 1 to reviewer Aiai**
>
> We thank reviewer Aiai for taking the time to review our work.
>
> **P1** _The idea novelty is limited. The specialization and orthogonalization idea are not that new._
>
> We thank the reviewer for raising this question, however, we would like to argue that our treatment of specialization and orthogonalization is novel and brings a significant contribution to the literature. We split our discussion into two key aspects of novelty: literature novelty and behavioral novelty. Before expanding on these points, we wish to make a small clarification. This reviewer mentioned that “the method works because intuitively it makes sense to enforce specialization of each unit and orthogonalization between units”. We wish to highlight that we do not enforce specialization and orthogonalization between the units themselves (as is well studied in the disentanglement literature in e.g. [1]), instead, **we enforce this behavior on the input attributions with respect to the hidden units**. We note that these two ideas are quite distinct.
>
> **Literature novelty** - Here we refer to the novelty of our method with respect to existing works in the literature. A number of previous works have considered using gradient attributions to improve model robustness to input perturbations (e.g. [2]) which is a useful defense against some adversarial attacks [3]. Typically, these methods use gradients with respect to the output layer rather than a hidden layer, in contrast to this work, as they wish to permute inputs while minimizing or maximizing the change in target. Furthermore, **invariance to small perturbations of inputs is often less desirable in the tabular setting** than in e.g. images, as one pixel is unlikely to change the class of an image while one important feature can significantly change the classification of a patient record or a credit decision. Another line of research uses input gradients to manually enforce a network to attend to human-crafted regions in the input (e.g. [4,5]). **This requires a manually annotated region for the network to attend**, which is clearly unrelated to the method presented in TANGOS. Finally, [6] considers using gradient attributions as priors to encourage certain properties such as smoothness between pixels in an image. Again, this work is in the context of **gradients with respect to the output layer** and therefore **does not consider the relationship between gradient attributions of hidden units**. We hope that this summary further demonstrates the novelty of this work with respect to existing literature. If the reviewer is aware of any specific existing works that deal with specialization and orthogonalization between the gradient attributions of latent units in neural networks, we would be happy to reevaluate our position on this issue.
>
> **Behavioral novelty** - Here we refer to the novel behavior induced in the network when compared to existing models. TANGOS regularization results in a model that **behaves very differently to existing methods of training**. In section 4.1 we showed that TANGOS does indeed result in models that attend to specialized and orthogonal regions in the input space as desired. This was in contrast to existing methods which achieve improved performance by other means. In section 4.2 we applied simple ensemble theory to show that TANGOS regularization results in a model in which the latent units behave more like a functioning ensemble. Again, this was in contrast to existing regularization methods. In addition to these points, we have added a **new experiment in Appendix H in which we perform an illustrative analysis of exactly how a TANGOS regularized model is distinct from standard training**. In order to maximize the interpretability of this experiment, we evaluate TANGOS on a simple image classification task with the MNIST dataset. By comparing the gradient attributions of the TANGOS-trained model with those of a baseline, we highlight that our method results in specialized and de-correlated latent units. We also find that in many cases we can provide a clear interpretation of the roles played by each latent unit, which provides an exciting direction for future research.
>
> To summarize, we believe this work to be novel with respect to existing literature as well as the behavior it induces in the network. We thank this reviewer and **we have added some minor clarification to reflect this discussion in the related works section of the main text**.
>
> _references included in next response due to character constraints_

---

> ### Author Response · Authors · 2022-11-14
> **Response 2 to reviewer Aiai**
>
> **P2** _There is no ablation study on which part is more important, specialization or orthogonalization._
>
> Thank you for raising this suggestion. As our current results in Figure 3 indicate, specialization and orthogonalization play different roles that together induce desirable behavior in the neural network. To address the reviewer's comment, we consider the performance gain due to joint regularization effects over applying each regularizer independently as **a new experiment in Appendix G**.
>
> This includes three separate settings: 1) when the specialization regularizer is applied independently (SpecOnly), here we set $\lambda_2=0$ and search over $\lambda_1 \in \{1, 10, 100\}$; 2) when the orthogonalization is applied separately (OrthOnly), we set $\lambda_1=0$ and search over $\lambda_2\in\{0.1, 1\}$; and lastly 3) when both are applied jointly (TANGOS), i.e. search over $\lambda_1\in \{1, 10, 100\}$ and $\lambda_2 \in \{0.1, 1\}$. We report the result of the ablation study in the table below, additionally adding the results to Appendix G.
>
> [[Additional results] Ablation study to investigate joint effects of specialization and orthogonalization.](https://i.imgur.com/c4TAqOM.png)
>
> Empirically, **we observe that the joint effects of both regularizers (i.e. TANGOS) is crucial to achieving consistently good performance**. On 5/6 datasets we evaluated on, jointly employing both regularizers outperformed applying either specialization or orthogonalization. Combining this with the results we have already laid out in Section 3.3 and Section 4.1, we hypothesize that this behavior has a simple explanation. Applying only specialization regularization, regardless of diversity, can inadvertently force the neurons to attend to overlapping regions in the input space. Correspondingly, simply enforcing orthogonalization, with no regard for sparsity, will result in neurons attending to non-overlapping yet spurious regions in the input. Thus, **we conclude that the two regularizers have distinct, but complementary, effects** that work together to achieve the desired regularization effect.
>
>
> [1] Higgins, Irina, et al. "beta-vae: Learning basic visual concepts with a constrained variational framework." (2016).
>
> [2] Drucker, H. and Le Cun, Y. (1992). Improving generalization performance using double backpropa- gation. IEEE Transactions on Neural Networks, 3(6):991–997.
>
> [3] Moosavi-Dezfooli, S.-M., Fawzi, A., Uesato, J., and Frossard, P. (2019). Robustness via curvature regularization, and vice versa. In Proceedings of the IEEE/CVF Conference on Computer Vision and Pattern Recognition, pages 9078–9086.
>
> [4] Liu, F. and Avci, B. (2019). Incorporating priors with feature attribution on text classification.
>
> [5] Chen, J., Wu, X., Rastogi, V., Liang, Y., and Jha, S. (2019). Robust attribution regularization. Advances in Neural Information Processing Systems, 32.
>
> [6] Erion, G., Janizek, J. D., Sturmfels, P., Lundberg, S. M., and Lee, S.-I. (2021). Improving performance of deep learning models with axiomatic attribution priors and expected gradients. Nature machine intelligence, 3(7):620–631.

---

> ### Author Response · Authors · 2022-11-17
> **Follow up**
>
> We wish to follow up to this reviewer with a further update to the manuscript.
>
> We have added an appropriate statistical test to determine whether the magnitude of improvement achieved by TANGOS is statistically significant. This is described as follows: We wish to perform a statistical test to determine if the population differences in performance paired at the dataset level are significant. As we do not wish to make any assumptions on the distribution of the losses, we seek a non-parametric version of the paired t-test. Therefore, the Wilcoxon signed-rank sum test [1] is appropriate, which provides **a non-parametric test of model performance across multiple datasets** [2]. We compare TANGOS to the best-performing baseline method (L2) as a one-tailed test to determine if the former's improvement is statistically significant.  We perform the test on both the regression and classification results reported in Table 1 and obtain p-values of 0.006 and 0.026 respectively, which can be interpreted as strong evidence to suggest the difference is statistically significant in both cases. **We have also included these results in the latest version of the paper on openreview**.
>
> **As the rebuttal period draws to a close, if we have addressed the points raised by this reviewer we hope that they will consider increasing their scores. Otherwise, we would be happy to respond to any further questions**.
>
> [1] Wilcoxon, Frank. "Individual comparisons by ranking methods." Breakthroughs in statistics. Springer, New York, NY, 1992. 196-202.
> [2] Demšar, Janez. "Statistical comparisons of classifiers over multiple data sets." The Journal of Machine learning research 7 (2006): 1-30.

---

### Official Review · Reviewer_xMqZ · 2022-11-04

**Confidence:** 4
**Correctness:** 4
**Technical Novelty And Significance:** 3
**Empirical Novelty And Significance:** 3
**Recommendation:** 6

**Clarity, Quality, Novelty And Reproducibility:**

The quality of the empirical evaluation, the clarity are all good.  The work appears reproducible.  The novelty is good, as I don't think any other work has tried to incorporate ensemble learning into neural nets in this way (via gradient attribution).

**Strength And Weaknesses:**


 Strengths
- I really like the overall direction of drawing upon the ensemble learning literature to help improve the performance of neural networks in tabular learning.  GBDTs are still SOTA in tabular learning, so taking the strengths of GBDTs and incorporating them into deep models is promising.
- The paper is very clear and the experimental setups/plots are all easy to follow
- When combined with other regularizers, TANGO appears to produce strong results across datasets
- The paper makes a strong case that the regularizer is both doing something unique vs other regularizers and that it accomplishes the ideas of sparsifying the diversifying the “weak learners”

Weaknesses
- The paper lacks comparison to GBDTs.  Given that GBDTs are still SOTA in the tabular setting, it’s important to compare against them.  Otherwise why do deep learning at all?
- The grid search for TANGO is larger than for the others with 6 choices, while for L1/L2 it’s 3 choices, dropout it’s 2 choices, and input noise it is 2 choices.  Maybe a slightly unfair comparison?


**Summary Of The Paper:**

This paper draws upon ideas from the ensemble learning and gradient attribution literatures to produce a neural network based approach to tabular learning.  The main ideas are as follows (1) each penultimate neuron should be sensitive to only a few of the input features (2) each penultimate neuron should be sensitive to non-overlapping sets of features wrt the other penultimate neurons.  The justification for this intuition is that in ensemble learning, ensemble learners benefit from diversity in its weak learners–and we can view a neural network as an ensemble of weak learners where the weak learners are the neurons of the penultimate layer.

**Summary Of The Review:**

Overall, I like the ideas in this paper and they seem like a promising way to make deep learning work in the tabular setting.  However, there is no comparison to GBDTs, which is critical in the tabular setting (see https://deepai.org/publication/why-do-tree-based-models-still-outperform-deep-learning-on-tabular-data).  For now, my rating is a 5.5 pending an evaluation with GBDTs.  Now, TANGO does not necessarily have to outperform GBDTs, but I would like there to be a case made that TANGO is a valuable step towards beating GBDTs.

===========I raise my score to a 6 after rebuttal.  If accepted, I recommend the authors spend some time to rewrite the paper such that the comparison with boosting methods is made more clear (i.e. why would a deep learning method that doesn't have pure generalization outperformance vs boosting be a valuable contribution?).  The rebuttal period helped make this more clear to me and I think the ideas in the paper are worthy of presentation at the conference.  One additional thing I might want to see in this paper is a proof-of-concept of TANGOS in multi-modal/meta/interpretability, though I leave it up to the authors.===========

---

> ### Author Response · Authors · 2022-11-14
> **Response 1 to reviewer xMqZ**
>
> We thank reviewer xMqZ for taking the time to review our work.
>
> **P1** _The paper lacks comparison to GBDTs. Given that GBDTs are still SOTA in the tabular setting, it’s important to compare against them. Otherwise why do deep learning at all?_
>
> We agree with this reviewer that a discussion of and comparison to GBDTs would make for a valuable addition to this work. We have therefore **added an additional experiment in Appendix I** in which we apply TANGOS to recent works which **assess deep learning methods capacity to compete with GBDTs on tabular data**. We describe this experiment and our findings in the next paragraph, but prior to this, we wish to set some context for this comparison. We do not claim the methods presented in this work to be a panacea for deep learning which render all non-neural methods redundant. Indeed, we agree with this reviewer that GBDTs can still be considered state-of-the-art in this domain due to their impressive comparative performance in addition to other factors such as computational efficiency and strong out-of-the-box performance. However, we also believe that further progress in applying deep learning for tabular data is an extremely promising research direction. Impressive progress has been made in recent years on this front, and the gap continues to reduce in terms of generalization performance (which we hope to contribute to in this work). Furthermore, **deep learning provides a number of promising advantages outside of raw performance** including multi-modal learning, meta-learning and certain interpretability methods to name just a few. For this reason, **we believe that progress in deep learning for tabular data in parallel to non-neural methods to be a valuable and worthwhile research direction**. We believe that TANGOS is an important contribution to this research direction providing improved performance as well as useful insights into why certain modelling behaviors may be particularly well suited to tabular data.
>
> In this paragraph, **we summarise the new experiment in Appendix I comparing TANGOS to GBDTs** in the setting of recently proposed architectures for tabular data. Specifically, we use the architecture proposed in [1] which combines a Feature Tokenizer which transforms features into embeddings with a multi-layer Transformer. We train this model with and without TANGOS regularization and compare to the performance of both XGBoost and CatBoost on two benchmark datasets from that work. While we do find that TANGOS improves performance we find that the boosting methods still perform best. We include a summary of the results in [this table](https://imgur.com/a/vpyIIbG) and **a complete description of the experiment in Appendix I**.
>
> [1] Yury Gorishniy, Ivan Rubachev, Valentin Khrulkov, and Artem Babenko. Revisiting deep learning models for tabular data. Advances in Neural Information Processing Systems, 34:18932–18943, 2021.

---

> > ### Comment · Reviewer_xMqZ · 2022-11-22
> > **Appendix I experiments perhaps incomplete?**
> >
> > For the experiment in Appendix I to be complete, I think you need two more things:
> >
> > (1) Results when using "benchmark regularizers"
> >
> > (2) Results when combining "benchmark regularizers" with TANGOS
> >
> > The first is valuable to see the improvement over benchmarks.  The second is important because a core message of the paper is the value of TANGOS as part of a combination of regularizers.  This is also a more general comment on the paper, but it is odd that in a tabular ML paper GBDTs are not a central part of the discussion and only relegated to the appendix.  GBDTs are the SOTA in tabular ML so they are really the core benchmark.

---

> > > ### Author Response · Authors · 2022-11-25
> > > **Update**
> > >
> > > We thank this reviewer for their continued engagement.
> > >
> > > Again, we agree that this is a fair point and have now run further evaluations in this section. We summarize these new results in what follows and include the complete updated text of Appendix I in the comment that follow. We would be happy to include these changes in a camera ready version of this paper.
> > >
> > > We now consider point (1) where we jointly optimize the Transformer along with all baseline regularizers (“tuned” and “baseline”)  and point (2) where we jointly optimize the Transformer along with the baseline regularizers and TANGOS (“tuned” and “+ TANGOS”). The [updated Table 9 is linked](https://imgur.com/a/LupXtEr) where the new results are included under the setting “Tuned”. In line with section D.2 of [1], ten iterations of random search tuning was sufficient with the hyperparameters achieving the best validation performance selected. We evaluate this model training over three seeds and perform their final evaluations on a held-out test set. This entire protocol is followed both with and without TANGOS in the search space (denoted “baseline” and “+ TANGOS”). The tuned boosting methods are also included for reference. **Again we find that optimizing TANGOS in-tandem with existing regularization in this state-of-the-art architecture results in additional performance gains**.
> > >
> > > _This is also a more general comment on the paper, but it is odd that in a tabular ML paper GBDTs are not a central part of the discussion and only relegated to the appendix._
> > >
> > > Our focus in this work was improving deep learning methods for tabular data. We agree that including boosting in this conversation improves the paper as a whole. We also believe that there is an over emphasis on comparing neural networks to boosting in the literature. Both methods have inherent advantages and disadvantages and are worthy of research in their own right (as discussed in our previous comment and Appendix I). However, recent deep learning methods for tabular data generally center their contribution on outperforming boosting which fails to replicate when evaluated on independent benchmarks [2]. Instead, we have presented a well-motivated method which performs well and provides a promising foundation for future work. That said, **we do agree that a discussion on boosting should also be a primary part of this conversation and not just in the appendix**. Therefore we propose to include the main results from this section in the main paper as a third experiment in our experiments section (specifically this would become Section 5.3). Unfortunately, ICLR does not provide additional space in the camera-ready version of the paper. However, we believe that by moving a portion of the related works to the appendix we can provide sufficient space to include the key idea and results of this experiment with the finer details of the experimental design remaining in the appendix. We also propose to extend our discussion on the relationship between deep learning and non-neural methods in this section. This extended discussion is included in our updated version of what is currently appendix I and is included in the comment that follow.
> > >
> > > [1] Gorishniy, Yury, et al. "Revisiting deep learning models for tabular data." Advances in Neural Information Processing Systems 34 (2021)
> > >
> > > [2] Shwartz-Ziv, Ravid, and Amitai Armon. "Tabular data: Deep learning is not all you need." Information Fusion 81 (2022): 84-90.

---

> > > > ### Author Response · Authors · 2022-11-25
> > > > **Updates to Appendix I**
> > > >
> > > > **What follows is the updated text of Appendix I.**
> > > >
> > > > While non-neural methods such as XGBoost (Chen & Guestrin, 2016) and CatBoost (Prokhorenkova
> > > > et al., 2018) are still considered state-of-the-art for tabular data (Grinsztajn et al., 2022), much progress
> > > > has been made in recent years to close the gap. Furthermore, differing learning paradigms have
> > > > various strengths and weaknesses outside of maximizing generalization performance, which is often
> > > > a consideration in practical applications. While boosting methods boast excellent computational
> > > > efficiency and strong out-of-the-box performance, neural networks have unique utility in, for example,
> > > > multi-modal learning (Ramachandram & Taylor, 2017), meta-learning (Hospedales et al., 2021)
> > > > and certain interpretability methods (Zhang et al., 2021). In this section, we provide additional
> > > > experiments applying TANGOS to a state-of-the-art transformer architecture for tabular data proposed
> > > > in Gorishniy et al. (2021). Specifically, this architecture combines a Feature Tokenizer which
> > > > transforms features into embeddings with a multi-layer Transformer (Vaswani et al., 2017). We
> > > > compare this FT-Transformer architecture to boosting methods in the default setting where we
> > > > evaluate out-of-the-box performance and the tuned setting where we jointly optimize the Transformer
> > > > along with its baseline regularizers. We describe these two settings in more detail next.
> > > >
> > > > Default Setting. In this setting, we use a 3 layer Transformer with a 32 dimensional feature
> > > > embedding size and 4 attention heads. Following the original paper we use Reglu activations, a
> > > > hidden layer size of 43 corresponding to a ratio of $\frac{4}{3}$ with the embedding size, Kaiming initialization
> > > > (He et al., 2015), and AdamW optimizer (Loshchilov & Hutter, 2017). Finally, we apply a learning
> > > > rate of 0.001. We compare this architecture with and without TANGOS regularization applied which
> > > > we refer to as “Baseline” and “+ TANGOS” respectively. We set λ1 = 1 and λ2 = 0.01 which were
> > > > found to be reasonable default values for specialization and orthogonalization in our experiments in
> > > > Section 5.
> > > >
> > > > Tuned Setting. Here we apply ten iterations of random search tuning over the same hyperparameters as in the original work with those achieving the best validation performance selected. We then evaluate this combination by training over three seeds and perform their final evaluations on a held-out test set. We search using the same distributions as in the original work and consider the following ranges. L2 regularization $\in [1e-06, 1e-03]$, residual dropout $\in [0.0, 0.2]$, hidden layer dropout $\in [0.0, 0.5]$, attention dropout $\in [0.0, 0.5]$, hidden layer to feature embedding dimension ratio $\in [1.0, 3.0]$, embedding dimension $\in [16, 48]$, number of layers $\in [1, 3]$, learning rate $\in [1e-04, 1e-03]$. In the ``+ TANGOS'' setting we also include $\lambda_1 \in [0.001, 10]$ and $\lambda_2 \in [0.0001, 1]$ with a log uniform distribution. All remaining architecture choices are consistent with the default setting and the original work.
> > > >
> > > > We ran our experiments on the Jannis (Guyon et al., 2019) and Higgs (Baldi et al., 2014) datasets.
> > > > These are both classification datasets consisting of 83733 and 98050 examples respectively. These
> > > > datasets were selected as they represent a significant number of input examples along with a middling
> > > > number of input features relative to the other tabular datasets explored in this work (54 and 28
> > > > respectively). We follow the experimental protocol of the boosting comparison in Grinsztajn et al.
> > > > (2022) using the same training, validation and test splits and reporting mean test accuracy over three
> > > > runs. Therefore we obtain the same results for boosting as reported in that work.
> > > >
> > > > The results of this experiment are reported in Table 9 where we find that TANGOS does indeed
> > > > have a positive effect on the FT-Transformer performance. While we do not claim that TANGOS
> > > > regularization results in neural networks that outperform Boosting methods, these results indicate that
> > > > TANGOS regularization can contribute to closing the gap and may play a key role when combined
> > > > with other methods as highlighted in Kadra et al. (2021). We believe this to be an important area for
> > > > future research and, in particular, expect that architecture-specific developments of the ideas presented
> > > > in this work may provide further improvements on the results obtained in this section.

---

> > > > > ### Comment · Reviewer_xMqZ · 2022-11-28
> > > > > **Can you expand further on the utility of deep learning on tabular data?**
> > > > >
> > > > > For both me and any future readers of the paper, it would be good to expand further on "including multi-modal learning, meta-learning and certain interpretability methods to name just a few".  The value of TANGOS + deep learning over boosting is an existential question for this paper, so motivating this better will help me and other readers evaluate the contribution better.

---

> > > > > > ### Author Response · Authors · 2022-11-29
> > > > > > **Further details on this point**
> > > > > >
> > > > > > We thank the reviewer for their suggestion and we are happy to expand on this point. We agree that integrating this discussion into the main paper will provide clearer motivation for this work. We will make this motivation explicit by adding it to the introduction and expanding upon the specific examples in the Appendix with the following discussion:
> > > > > >
> > > > > > **Multi-modal learning** refers to the task of modelling data inputs that consist of multiple data modalities (e.g. image, text, tabular). As one might intuit, jointly modelling these multiple modalities can result in better performance than independently predicting from each of them [1,2]. Deep learning provides a uniquely natural method of combining modalities with the advantages of (1) modality-specific encoders (2) that are fused into a joint downstream representation and trained end-to-end with backpropagation, and (3) superior modelling performance in many modalities such as images and natural language. Healthcare is a domain in which multi-modal learning is particularly salient [3]. Recent work in [4] showed that jointly modelling tabular clinical records using an MLP with medical images using a CNN outperforms the non-multi-modal baselines. Elsewhere in [5], a multi-modal approach is taken in combining input modalities based on the preprocessing of functional magnetic resonance imaging and region of interest time series data for the diagnosis of autism spectrum disorder. A resnet-18 encodes one modality while an MLP encodes the other resulting in superior performance when analysed in an ablation study. In this setting, progress in modelling each of the individual modalities is likely to result in better performance of the system as a whole. Interestingly, [1] identified regularization techniques for improved cross-modality learning as an important research direction. We believe that further development of the ideas presented in this work could provide a powerful tool for balancing how models attend to multiple input modalities.
> > > > > >
> > > > > > **Meta-learning** aims to distil the experience of multiple learning episodes across a distribution of related tasks to improve learning performance on future tasks [6]. Deep learning-based approaches have seen great successes on this problem in a variety of fields. In the tabular domain, with careful consideration of the shared information between tasks, recent works have also shown promising results in this direction by developing methods for transferring deep tabular models across tables [7,8]. In particular, in [8] it was noted that “representation learning with deep tabular models provides significant gains over strong GBDT baselines”, also finding that “the gains are especially pronounced in low data regimes”.
> > > > > >
> > > > > > **Interpretability** is an important area of deep learning research aiming to provide users with the ability to understand and reason about model outputs. Certain classes of interpretability methods have recently been developed that provide distinct forms of interpretability relying on the hidden representations of neural networks. In such models, probing the representation space of a deep model permits a new type of interpretation. For instance, [9] studies how human concepts are represented by deep classifiers. This makes it possible to analyze how the classes predicted by the model relate to human understandable concepts. For example, one can verify if the stripe concept is relevant for a CNN classifier to identify a zebra, as demonstrated in the paper. Another example is [10], which proposes to explain a given example with reference to a freely selected set of other examples (potentially from the same dataset). A user study was carried out in this work which concluded that, among non-technical users, this method of explanation does affect their confidence in the model’s prediction. These powerful methods crucially rely on the model's representation space, which effectively assumes that the model is a deep neural network.
> > > > > >
> > > > > >
> > > > > > _[references in next comment]_

---

> > > > > > ### Author Response · Authors · 2022-11-29
> > > > > > **References**
> > > > > >
> > > > > > _[continued from previous comment]_
> > > > > >
> > > > > > [1] Ramachandram, Dhanesh, and Graham W. Taylor. "Deep multimodal learning: A survey on recent advances and trends." IEEE signal processing magazine 34.6 (2017): 96-108.
> > > > > >
> > > > > > [2] Guo, Wenzhong, Jianwen Wang, and Shiping Wang. "Deep multimodal representation learning: A survey." IEEE Access 7 (2019): 63373-63394.
> > > > > >
> > > > > > [3] Acosta, Julián N., et al. "Multimodal biomedical AI." Nature Medicine 28.9 (2022): 1773-1784.
> > > > > >
> > > > > > [4] Wu, Xinglong, et al. "Deep multimodal learning for lymph node metastasis prediction of primary thyroid cancer." Physics in Medicine & Biology 67.3 (2022): 035008.
> > > > > >
> > > > > > [5] Tang, Michelle, et al. "Deep multimodal learning for the diagnosis of autism spectrum disorder." Journal of Imaging 6.6 (2020): 47.
> > > > > >
> > > > > > [6] Hospedales, Timothy, et al. "Meta-learning in neural networks: A survey." IEEE transactions on pattern analysis and machine intelligence 44.9 (2021): 5149-5169.
> > > > > >
> > > > > > [7] Zifeng Wang and Jimeng Sun. Transtab: Learning transferable tabular transformers across tables. In Advances in Neural Information Processing Systems, 2022.
> > > > > >
> > > > > > [8] Levin, Roman, et al. "Transfer Learning with Deep Tabular Models." arXiv preprint arXiv:2206.15306 (2022).
> > > > > >
> > > > > > [9] Kim, Been, et al. "Interpretability beyond feature attribution: Quantitative testing with concept activation vectors (tcav)." International conference on machine learning. PMLR, 2018.
> > > > > >
> > > > > > [10] Crabbé, Jonathan, et al. "Explaining Latent Representations with a Corpus of Examples." Advances in Neural Information Processing Systems 34 (2021): 12154-12166.

---

> ### Author Response · Authors · 2022-11-14
> **Response 2 to reviewer xMqZ**
>
> **P2** _The grid search for TANGO is larger than for the others with 6 choices, while for L1/L2 it’s 3 choices, dropout it’s 2 choices, and input noise it is 2 choices. Maybe a slightly unfair comparison?_
>
> **Edit (17/11)**: We have now included extra runs and updated the manuscript, please see [further updates](https://openreview.net/forum?id=n6H86gW8u0d&noteId=HNu3oy6mgff) comment for details.
>
> We thank the reviewer for raising this issue and agree that, at first, this choice may appear to be an unfair comparison. However, this was a considered choice which we believe provided the most fair comparison. Let us explain our rationale:
>
> The primary factor in this decision was the size of the search space. TANGOS requires a search over two hyperparameters while the aforementioned baselines only require a search over a single dimension. Therefore, if we suppose there to be some optimal setting of hyperparameters uniformly distributed within this space, then the expected distance between a randomly selected search point and this optimal point is clearly greater in two dimensions than in one. Of course, we are not exactly in this scenario, however, it does illustrate the point that **searching in a two-dimensional space does require a greater number of search iterations for a fair comparison**.
>
> A second factor was the well-established search ranges of the baseline methods. Given the vast usage of these methods over many years of research, **we were provided with a strong prior on reasonable values to consider**. Indeed, the grid search values we considered matched those of previous works [2]. As it is a novel method, we did not have strong intuitions for a reasonable search range for TANGOS and based our choice on limited preliminary trials. For further insights on the interaction between the two regularization parameters controlling specialization and orthogonalization, we refer the reviewer to the **new ablation study provided in Appendix G**. There we find that neither factor is generally effective in isolation and, instead, finding a suitable interaction between the two is essential.
>
> Finally, we note that the final model selected is evaluated on independent test data for just one hyperparameter setting per dataset. We mention this only to confirm that **differing search space sizes do not result in extra evaluations on the test set**.
>
> To summarize, it is reasonable that a two-dimensional search space requires more evaluations than a one-dimensional search space. The exact best number is unclear and depends on a number of factors including the level of dependence between the two factors, our prior knowledge on the search space dynamics, and the sensitivity of the method to precise tuning. Given that the baselines considered up to three values, one could argue for any number of searches in the range [3^1, 3^2] for TANGOS. **We believe that 6 was a suitable compromise**, falling in the middle of that range.
>
> [2] Kyono, Trent, Yao Zhang, and Mihaela van der Schaar. "CASTLE: regularization via auxiliary causal graph discovery." Advances in Neural Information Processing Systems 33 (2020): 1501-1512.

---

> ### Author Response · Authors · 2022-11-17
> **Further updates**
>
> We wish to follow up to this reviewer with two further updates to the manuscript reflecting their feedback.
>
> (1) As a result of the discussion regarding the search space (P2), **we have extended our experiments with additional hyperparameters** considered for input noise and dropout. Consistent with our response from Monday, we now search over three hyperparameters for all suitable baseline methods. This results in some changes to the scores reported in Table 1 (**these changes have been added to the manuscript** and are summarized [in this link](https://imgur.com/a/4v2Fd1r)) where we still find TANGOS to be outstanding among the eight methods assessed.
>
> (2) We have also added an appropriate statistical test to determine whether the magnitude of improvement achieved by TANGOS is statistically significant. This is described as follows: We wish to perform a statistical test to determine if the population differences in performance paired at the dataset level are significant. As we do not wish to make any assumptions on the distribution of the losses, we seek a non-parametric version of the paired t-test. Therefore, the Wilcoxon signed-rank sum test [1] is appropriate, which provides **a non-parametric test of model performance across multiple datasets** [2]. We compare TANGOS to the best-performing baseline method (L2) as a one-tailed test to determine if the former's improvement is statistically significant.  We perform the test on both the regression and classification results reported in Table 1 and obtain p-values of 0.006 and 0.026 respectively, which can be interpreted as strong evidence to suggest the difference is statistically significant in both cases. **We have also included these results in the latest version of the paper on openreview**.
>
> **As the rebuttal period draws to a close, if we have addressed the points raised by this reviewer we hope that they will consider increasing their scores. Otherwise, we would be happy to respond to any further questions**.
>
> [1] Wilcoxon, Frank. "Individual comparisons by ranking methods." Breakthroughs in statistics. Springer, New York, NY, 1992. 196-202.
>
> [2] Demšar, Janez. "Statistical comparisons of classifiers over multiple data sets." The Journal of Machine learning research 7 (2006): 1-30.

---

### Author Response · Authors · 2022-11-18
**Summary of changes**

We thank the reviewers for investing their time in reviewing our work. We provide a summary of the changes made to the manuscript which we believe should thoroughly address the reviewers' queries.

_29/11_

* **Extended discussion on motivation in Introduction and Appendix I** - We have added a clarifying discussion on the motivation of deep learning methods for tabular data reflecting the inherent advantages and disadvantages, especially with respect to gradient boosting methods. These changes are reflected in our discussion with reviewer xMqZ [here](https://openreview.net/forum?id=n6H86gW8u0d&noteId=stRt5qJgv9) and [here](https://openreview.net/forum?id=n6H86gW8u0d&noteId=ahepLP6w-U).

_18/11_

* **Performance with increasing data experiment added in Appendix J** - We empirically **validate TANGOS gains with increasing dataset size on the largest benchmark task** (416,188 examples) from the suite of tabular benchmarks proposed in [1]. We find that TANGOS achieves significantly better performance at all fractions of the data as summarized [in Figure 13](https://imgur.com/a/Bf86nlN).

_17/11_
- **Statistical test verifying results in Section 5.1** - We have added the results of a Wilcoxon signed-rank sum test providing strong evidence that **TANGOS gains are statistically significant**.
- **Extended hyperparameter search in Section 5.1** - We **extended the search space** of the appropriate baseline methods to ensure differences observed were not due to differences in the search space. The updated results in the manuscript can be seen [here](https://imgur.com/a/4v2Fd1r).

_14/11_
- **Behavior analysis experiment added in Appendix H** - We further validate that a TANGOS-trained model behaves in a novel manner. We also apply TANGOS on an **alternative architecture** (CNN). A side-by-side comparison of the attributions of a [TANGOS trained model beside a baseline model](https://imgur.com/a/WN759hz) highlights that **TANGOS regularization does indeed result in novel behavior**.
- **Ablation study added in Appendix G** - We perform an ablation study confirming that the **two regularization terms have distinct, but complementary, effects** that in combination achieve the desired regularization effect. The results are included [in Table 6](https://i.imgur.com/c4TAqOM).
- **Tabular architectures and boosting comparison experiment added in Appendix H** - We show that TANGOS is effective on a recently proposed **transformer architecture** for tabular data. We also provide evidence that TANGOS contributes to **closing the gap on gradient-boosting methods** for neural networks in terms of generalization performance. We include these results [in Table 9](https://imgur.com/a/LupXtEr) (**Edit**: This experiment was extended with additional tuned runs following discussion with xMqZ).
- **Related works Section 2** - We made some minor updates in the related works section to highlight our novelty with respect to existing literature.


We believe that our rebuttal paired with these additional experiments and results should have addressed the reviewers' points, but we would be more than happy to clarify any residual concerns.

[1] Kadra, A., Lindauer, M., Hutter, F. and Grabocka, J., 2020. Regularization Cocktails for Tabular Datasets.

---

### Decision · Program_Chairs · 2023-01-20

**Decision:**

Accept: poster

**Justification For Why Not Higher Score:**

Small gains when compared to baselines and lack of enough error analysis.

**Justification For Why Not Lower Score:**

Well written paper on a relatively less studied but very important domain. Novel ideal and conceptually simple idea. Extensive experiments. Theoretical motivation.

**Metareview: Summary, Strengths And Weaknesses:**

This well-written paper proposes two conceptually simple regularizers for neural networks for tabular data. These regularizers promotes that the final hidden features is strongly influenced by a single unique input feature dimension. Ideas reads very similar to the disentangling representation learning in generative model literature. Authors motivate their method using theory of ensembles. Authors show extensive experiments to argue that their regularizer performs better than baseline regularizers, and it even improves the performance in-tandem with other regularizers. These ideas could be useful for future research and even outside tabular domain. Authors provided many experimental results, engaged with the reviewers, and considerably improved the paper after reviewer feedback.

A lot of results lack/initially lacked proper presentation of error/deviations. For example, why isn't the error bars/standard errors directly given in Table 1 andd Figure 5. It is also not explicitly mentioned that most gains in these results are within the error margin, which could be misleading to the readers. It is not clear how newly given Wilcoxon test was performed or whether it is appropriate considering the error bars. I got a different resullt for L2 - TANGOS Regression in Table1:  `scipy.stats.wilcoxon([0.032, 0.166, 0.093, 0.244, 0.637, 0.387, 1.276, 0.573, 0.382, 0.325], y=[0.029, 0.183, 0.099, 0.277, 0.644, 0.411, 1.274, 0.580, 0.418, 0.332])=WilcoxonResult(statistic=3.0, pvalue=0.012287335657219152)`.

AC notes that the updated text given in https://openreview.net/forum?id=n6H86gW8u0d&noteId=stRt5qJgv9 doesn't match the pdf currently (see Tuned setting). Therefore, it is assumed that the authors will correct this error in the next revision. Further, it is also assumed that the authors will add discussion about GBDTs to the main text, as proposed. It is not clear why "Interpretability" cannot be achieved using non-DL methods like GBDT, questioning the motivation for this paper.

**Note From Pc:**

if the above contains the word "oral" or "spotlight" please see: "oral" presentation means -> notable-top-5% and "spotlight" means -> notable-top-25%. As stated in our emails, we are disassociating presentation type from AC recommendations

---

> ### Author Response · Authors · 2023-02-23
> **Camera ready updates made**
>
> We thank the AC and three reviewers for their careful consideration of our work.
>
> Briefly responding to the minor points raised in the AC's comment, the camera-ready version now (1) explicitly mentions that several results have overlapping error intervals, (2) includes the results comparing to GBDTs with reference to the extended discussion in the Appendix, and (3) has synchronized the linked comment to the text in the paper (this difference was only due to the comment being made after the deadline for openreview to upload updates to the pdf).
>
> Finally, we followed the procedure described in [1] for a one-sided Wilcoxon signed-rank sum test. The small difference can be attributed to parameter settings in the scipy implementation. Identical p-values may be obtained by running
> ```scipy.stats.wilcoxon([0.032, 0.166, 0.093, 0.244, 0.637, 0.387, 1.276, 0.573, 0.382, 0.325], y=[0.029, 0.183, 0.099, 0.277, 0.644, 0.411, 1.274, 0.580, 0.418, 0.332], alternative='less', method='approx') = WilcoxonResult(statistic=3.0, pvalue=0.006143667828609576) ```
>
>
> [1] Demšar, Janez. "Statistical comparisons of classifiers over multiple data sets." The Journal of Machine learning research 7 (2006): 1-30.